# "Pochva": a new hydro-thermal process model in soil, snow, vegetation for application in atmosphere numerical models

Oxana Drofa[1]

[1]Istitute of Atmospheric and Climate of National Research Council (CNR-ISAC), Bologna, 40129, Italy

*Correspondence to*: Oxana Drofa (o.drofa@isac.cnr.it)

**Abstract**. This work presents the land model "Pochva". "Pochva" is a model of hydro-thermal processes at the Earth surface and in the underlying medium. The model simulates the main hydro-thermal parameters of the surface, soil layer, vegetation and snow layer. Its soil process scheme allows to use physical parameters having vertical variations along the soil profile. Its snow process scheme is a multilayer scheme and has a numerical algorithm that allows solving both extremely thin and
extremely thick layer cases. The model is characterized by a particular accuracy in simulating the water phase transitions in soil and snow, and by the autonomy in the determination of the lower boundary condition in the soil column. The model can be used as a stand-alone land-surface model driven by observed or analytical forcing data, or coupled to an atmospheric model, either global or limited-area, either in forecast regime or climatic (hindcast) regime. The results of coupling "Pochva" to the numerical weather prediction limited-area model "Bolam" are presented in this article.

## 1 Introduction

Water mass balance and energy balance at the Earth surface are key processes in a numerical model of the atmosphere. These processes determine the condition at the lower boundary for the main atmospheric parameters as well as the air parameters in the surface layer. The surface hydro-thermal conditions are simulated by a scheme (or model) of hydro-thermal processes in
the surface and in the underlying media, composed by the soil layer, possibly covered by vegetation, and by the snow layer. The simulation of hydro-thermal processes in the underlying media in the current models has evolved significantly from a simple soil scheme (e.g. Deardorff, 1978) to complex vegetation structures with multilayer soil hydrology and energy and multilayer snow. Examples of currently used land surface schemes include the Interaction Soil-Biosphere-Atmosphere model (ISBA, Noilhan and Planton, 1989); the Canadian Land Surface Scheme (CLASS, Verseghy, 1991; Verseghy et al., 1993); the
Tiled ECMWF Scheme for Surface Exchanges over Land model (TESSEL, Viterbo and Beljaars, 1995), including multilayer snow scheme (Arduini et al., 2019), the NOAH model (Ek et al., 2003); the Common Land Model of National Center of Atmospheric Research (USA) and Sun Yat-sen University (China) (Dai et al., 2003); the Community Land Model (CLM, Oleson et al., 2010), Joint UK Land Environment Simulator (JULES, Best et al., 2011), GEOtop (S. Endrizzi et al., 2014). The

important role of correctly simulating the interaction between the atmosphere and the land surface for current atmospheric and

climate research is discussed in a clear and complete manner in (Santanello et al., 2018).

The large number and variety of existing models is due to the fact that different models underline different processes in soil and vegetation. The differences are related to the different purposes of their application: weather prediction, study of atmospheric processes, simulation of snow cover and avalanche prediction, climate simulations coupled with biosphere models. Some models pay particular attention to hydro-thermal exchange processes in the soil, take into account phase

transition processes in soil water, and processes in the snow layer, they accurately describe the fluxes at soil surface. These models are more suitable for application in modelling of atmospheric processes and weather forecasting. Other models pay more attention to an accurate description of the processes related to vegetation, distinguishing high and low vegetation, simulating in detail processes like evapotranspiration and providing an accurate simulation of carbon cycle. These models are more suited to climate and Earth system modelling.

The model proposed in the present work is closer to the first class of models, i.e. is more suitable for models targeted at the study of atmospheric process and weather prediction models. In the proposed model, special attention was paid to the accuracy of description of heat and moisture exchange in soil, including water phase transitions, to the accuracy and reliability in the description of processes in the snow layer, including water melting and re-freezing, to the inhomogeneity in soil parameters along the vertical, to the definition of thermal and hydraulic conductivity in the different situations, to the problem of defining

the atmospheric humidity at the contact surface with the soil and vegetation leaves, to the problem of determining the albedo and density of snow cover depending on its history. In the proposed model an original method is employed for defining bottom boundary conditions for temperature and humidity, making the model autonomous within the context of a known climatology or climatological drift. The model can also be useful for the simulation of snow cover for avalanche prediction purposes since the snow module is independent of the other modules and can be applied separately. The model can also be used with a forcing

derived from observational data for defining the fluxes between the surface and the atmosphere as well as for idealised column simulations.

The present paper is divided into eight sections, the second, third, fourth, fifth and sixth are devoted to the description of the schemes included in the model: surface processes, processes in the vegetation, heat exchange processes in the soil, moisture exchange processes in the soil, processes in the snow layer. The seventh section contains the description of the numerical

experiments and their results. The last section contains the conclusions and some discussion about the critical points of the model and of the verification results.

**2 Surface processes scheme**

The interaction between the atmosphere and the Earth surface from the point of view of atmospheric modelling, occurs mainly by means of fluxes of heat and moisture. The main parameter describing the thermal state of the soil environment in the present

model is the entropy. This variable has been chosen since it simplifies the description of the water phase changes following the idea proposed in (Pressman, 1994).

The state of the atmosphere in interaction with the Earth surface is described by air temperature, specific humidity and pressure at the lowest atmospheric level, turbulent transfer coefficients between surface and lowest atmospheric level, total net radiation flux, fluxes of atmospheric precipitation in liquid and crystal phases. The thermodynamic state of the surface is described by

two parameters, temperature and specific humidity of the air at the surface. The values of these two parameters depend on the whole state of the underlying surface.

## 2.1 State of the soil surface

According to the principles adopted in this work, a unit size of underlying surface is composed of a set of fractions each of which exhibits uniform characteristics from the point of view of the interaction with the atmosphere. The soil surface can be

covered by vegetation (grass, shrub), high vegetation (trees, woodland) and snow. Snow cover and high vegetation imply the presence of a particular layer with its own thermodynamic characteristics, thus with a distinct temperature. For this reason, it is not possible to consider them as a fraction of the surface of the soil itself, but it is necessary to consider them as independent "columns". As a consequence, if we introduce the concept of fraction of snow cover, it is necessary to divide the surface into three independent columns, each having its own temperature even in case of equal upper and lower boundary conditions. In

the present version of the model here described, however, the following simplifying assumptions have been made: the vegetation has not been divided into high and low and the two types, possibly mixed, are considered as a part of the soil surface with particular characteristics; the snow layer can either cover all the surface or not exist at all, a fractional snow cover is introduced only as a diagnostic field to allow computation of the radiative characteristics of the surface (albedo, emissivity). Under these assumptions, the soil surface can either consist of bare soil possibly partly covered by low vegetation, which in

turn may be partly covered by water, or consist of snow cover. These two states of the surface can transit into one another but cannot exist simultaneously.

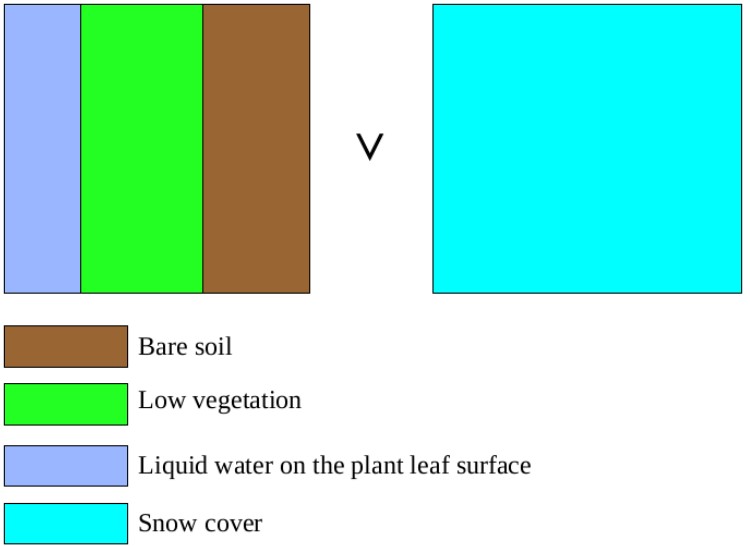

☐ Bare soil

☐ Low vegetation

☐ Liquid water on the plant leaf surface

☐ Snow cover

**Figure 1: Scheme of the soil surface.**

In the area of a single model grid cell the vegetation fraction is determined using an external vegetation parameter dataset. The fraction of vegetation leaves covered by water is determined by the ratio between the water mass deposited on the leaf surface and the maximum value of this mass, which is also evaluated using a dataset.

In this way, the surface interacting with atmosphere can be of the following types:

- without snow cover ($F_{snow}=0$):
  1) $F_{baresoil} = 1 - F_{veg}$ bare soil

  2) $F_{veg}^{dry} = F_{veg} \cdot \left(1 - F_{veg}^{leaf\ wet}\right)$ vegetation leaves not covered by water

  3) $F_{veg}^{wet} = F_{veg} \cdot F_{veg}^{leaf\ wet}$ vegetation leaves covered by water

- in case of snow cover, under the indicated assumptions the only type of interacting surface is:
  4) $F_{snow}=1$.

More precisely, in presence of snow, we always assume $F_{snow}=1$ for the computation of moisture fluxes, while for the entropy fluxes, $F_{snow}=1$ holds only if the snow cover has a minimal thickness, otherwise, for a shallow snow layer, entropy fluxes are computed under the assumption of $F_{snow}=0$ (see the explanation of the snow scheme for more details).

For each of these four surface types, surface air temperature and humidity have to be defined. The overall surface air temperature and humidity for the whole grid cell are then computed as an average of these values weighted with the fractional area of each surface type.

**2.2 Air temperature and humidity on bare soil**

For bare soil, the surface air temperature ($T_{surf\ soil}$) is equal to the temperature of the upper soil layer ($T_{soil0}$). The air specific humidity $q_{v\ surf\ soil}$ (kg kg-1) is defined according to the diagnostic expression

$q_{v\,surf\,soil} = q_{v\,atm\,1} \cdot (1 - \alpha_{soil}) + q_{v\,sat}(T_{soil\,0}) \cdot \alpha_{soil},$     (1)

where: $q_{v\,atm\,1}$ is the air specific humidity at the lowest atmospheric level (kg kg$^{-1}$), $q_{v\,sat}(T_{soil0})$ is the saturation air specific

humidity at temperature $T_{soil0}$ computed as:

$$q_{v\,sat} = \begin{cases} q_{v\,sat}^{water}(T_{soil0}),\ if\ T_{soil0} \geqslant T_0 \\ q_{v\,sat}^{ice}(T_{soil0}),\ if\ T_{soil0} < T_0 \end{cases},$$     (2)

where, in turn, $q_{v\,sat\,water}$ and $q_{v\,sat\,ice}$ are saturation air specific humidity over liquid water and ice respectively, $\alpha_{soil}$ is an

empirical coefficient, $To = 273.15$ (K) is constant.

For the definition of the empirical coefficient $\alpha_{soil}$ an original method is proposed in this work. The approach proposed has

been formulated following the method proposed in (Kondo et. Al, 1990) using both concepts of turbulent exchange and of

influence of top soil moisture to surface air humidity. Various empirical parameters have been introduced by the author and

presented in formulae (3), (4), (5). These formulations and the values of parameters are the results of many numerical

experiments and statistical verification on a big number of observational meteorological stations (see section 7). The

definition is the following:

$$\alpha_{soil} = \frac{2 \cdot F_2^{soil}}{e^{F_1^{soil}K_v^{turb}} + e^{-F_1^{soil}K_v^{turb}}},$$     (3)

where:

$K_v^{turb}$ is the coefficient of water vapour turbulent exchange in the lowest 1 m of the surface layer (m$^2$s$^{-1}$),

$F_1^{soil}$ and $F_2^{soil}$ are empirical functions of the relative moisture contents of the upper soil layer:

$$F_1^{soil} = 7 \cdot [2 + 3 \cdot (1 - q_0^{rel})^{0.2 + 0.05 \cdot b}],$$     (4)

$$F_2^{soil} = 1 - 0.8 \cdot (1 - q_0^{rel})^{0.2 + 0.05 \cdot b},$$     (5)

where b is the so-called"soil exponent" (Clapp and Hornberger, 1978), $q_0^{rel}$ is soil relative water content at the top level (see

section 4). The choice of the values defining the empirical function is crucial since it strongly influences the magnitude of the

water vapour flux and its associated latent heat flux, which, in turn, influences the soil surface temperature and the air

temperature at 2 m routinely used for the verification of numerical weather predictions. The methodology used for defining

these empirical coefficients and their influence on the numerical weather forecast will be treated in a separate publication

dedicated to the evaluation of the numerical model results.

**2.3 Air temperature and humidity over vegetation with leaves not covered by water**

The air temperature in this case ($T_{veg}^{dry}$) is equal to the temperature of the topmost soil layer ($T_{soil0}$). For the definition of air

humidity two cases are distinguished: the evapotranspiration is active or is not active. The conditions under which

evapotranspiration is or is not active are shown in section 3.

The method of numerical approximation of evapotranspiration presented here has been formulated using the approach

presented in (Pressman, 1994).

In the case of lack of evapotranspiration, the air humidity is equal to the air humidity at the lowest atmospheric layer while in the case of active evapotranspiration by leaves not covered by water it is defined analogously to the case of bare soil:

$$q_{v\,veg}^{dry} = \begin{cases} q_{v\,atm}, & if\ evapotranspiration\ not\ active \\ q_{v\,atm} \cdot (1 - \alpha_{veg} \cdot \beta_{veg}) + q_{v\,sat}(T_{soil\,0}) \cdot \alpha_{veg} \cdot \beta_{veg}, & if\ evapotranspiration\ active \end{cases} \tag{6}$$

where:

$q_{v\,veg}^{dry}$ is the air specific humidity over respiring plant leaf,

$q_{v\,atm}$ is air specific humidity at the bottom atmospheric level (kg kg$^{-1}$),

$q_{v\,sat}(T_{soil0})$ is saturation air specific humidity (kg kg$^{-1}$) at soil surface temperature $T_{soil0}$, (see paragraph 2.2),

$\alpha_{veg}$ is an empirical parameter depending on the magnitude of the evapotranspiration activity,

$\beta_{veg}$ is a parameter depending on the moisture content in the root layer of the soil, the definition of this parameter is given by equations 11-13 (see below).

The approximation of the parameter defining the evapotranspiration activity is similar to the method presented by equation (3). The parameter is given by:

$$\alpha_{veg} = \frac{2 \cdot F_2^{veg}}{e^{F_1^{veg} \cdot K_v^{turb}} + e^{-F_1^{veg} \cdot K_v^{turb}}}, \tag{7}$$

where $F_1^{veg}$ and $F_2^{veg}$ are empirical functions depending on the turbulent exchange coefficients for water vapour in the lowest 1 m of the atmospheric surface layer and on the intensity of the evapotranspiration process which in turn depends on the flux

of incoming visible solar radiation and on LAI (Leaf Area Index) following the conception proposed in (Viterbo et. al, 1995) according to:

$$F_1^{veg} = 30 \cdot \left[ -1.9 \cdot \frac{LAI}{LAI_{max}} + 2 \right], \tag{8}$$

$$F_2^{veg} = min \left[ \left[ min \left( \frac{F_{rad\,vis}}{600}, 1 \right) \right]^{0.3}, \left( \frac{LAI}{LAI_{max}} \right)^{0.2} \right], \tag{9}$$

where

$F_{rad\,vis}$ is the flux of visible solar radiation at the surface (W m$^{-2}$),

$LAI_{max}$ is the maximum value of LAI in the static global database used.

In this case, as in the case of air surface humidity over bare soil, the various empirical parameters in formulae (7), (8), (9) have been introduced by the author and their values have been obtained as result of many numerical experiments and statistical verification.

In order to evaluate the parameter $\beta_{veg}$, a description of the finite-difference representation of the vertical space coordinate used in the model is described here.

As a vertical coordinate, the geometrical length (depth) is used, with the origin positioned at the surface and values increasing with depth. The vertical computational domain is divided into full and half levels, with the upper full level having index zero, and with each half level having the same index as the full level located below it; the level indexes increase with depth (see fig.

175    2).

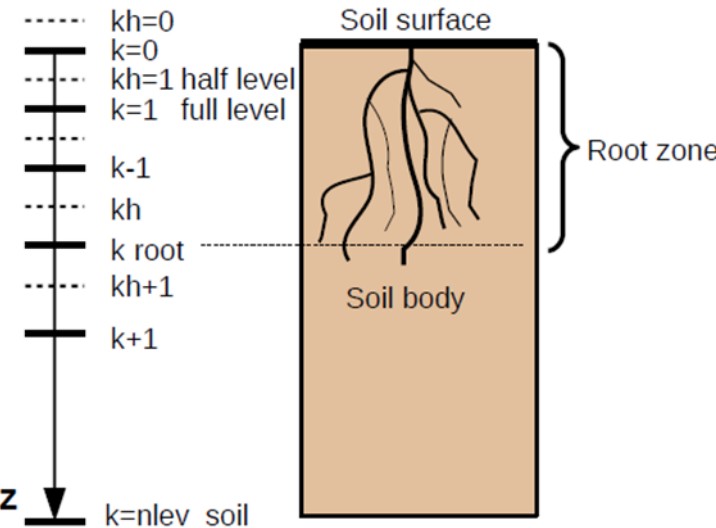

**Figure 2: Finite-difference discretisation of the vertical coordinate in the soil.**

The upper part of the soil column may contain plant roots. The depth of the root layer is defined as:

$$Z_{root} = \sum_{k=0}^{k_{root}} (z_{kh+1} - z_{kh}),$$  (10)

Where $z$ $(m)$ is the space coordinate in soil, $k$ is the index of full level in the vertical discretization, $k_{root}$ is the index of the deepest root zone level. The root zone depth is defined using a suitable vegetation dataset. In this work the distribution of vertical soil levels is not uniform, the thickness of soil layers increases with depth according to an exponential law, but it is possible to apply any other distribution. The depth of the soil bottom in the present scheme may be different for water exchange processes and for thermal exchange processes, depending on bottom boundary conditions, and they may differ from one geographical location to another according to local geographical characteristics.

$$\beta_{veg} = \frac{\sum_{k=0}^{k_{root}} (z_{kh+1} - z_{kh}) \cdot F(q_k^{rel})}{Z_{root}},$$  (11)

$$F(q_k^{rel}) = \begin{cases} 1, & if\ q_k^{rel} \geqslant q_k^{rel\,ref} \\ \dfrac{q_k^{rel} - q_k^{rel\,wilt}}{q_k^{rel\,ref} - q_k^{rel}}, & if\ q_k^{rel\,wilt} < q_k^{rel} < q_k^{rel\,ref}, \\ 0, & if\ q_k^{rel} \leqslant q_k^{rel\,wilt} \end{cases}$$  (12)

$$q_k^{rel} = \frac{q_k - q_k^{min}}{q_k^{max} - q_k^{min}},\ q_k^{rel\,wilt} = \frac{q_k^{wilt} - q_k^{min}}{q_k^{max} - q_k^{min}},\ q_k^{rel\,ref} = \frac{q_k^{ref} - q_k^{min}}{q_k^{max} - q_k^{min}},$$  (13)

where

$q_k$ and $q_k^{rel}$ are the soil specific volumetric (m$^3$ m$^{-3}$) and relative (proportion) water content  at level $k$,

$q_k^{max}$ and $q_k^{min}$ are the maximum and minimum soil specific volumetric contents at level $k$ (m$^3$ m$^{-3}$),

$q_k^{wilt}$ and $q_k^{rel\,wilt}$ are soil specific volumetric (m$^3$ m$^{-3}$) and relative (proportion) water contents at level $k$ at wilting point, i.e. at the water content at which plant evapotranspiration stops because of too dry soil,

$q_k^{\,ref}$ and $q_k^{rel\,ref}$ are soil specific volumetric (m$^3$ m$^{-3}$) and relative (proportion) water contents at level $k$ at "reference point", i.e. at the water content level at which plant evapotranspiration stops increasing because of highly wet soil.

## 2.4 Air temperature and humidity over vegetation with leaves covered by water

The air temperature in this case ($T_{veg}^{wet}$) is equal to the temperature of the topmost soil layer ($T_{soil0}$). The air humidity is equal to the saturation humidity at the given temperature:

$$q_{v\,veg}^{wet} = q_{v\,sat}(T_{soil\,0}), \tag{14}$$

where $q_{v\,sat}(T_{soil0})$ is the saturation air specific humidity (kg kg-1) at soil surface temperature $T_{soil0}$ (see paragraph 2.2).

## 2.5 Air temperature and humidity over snow cover

The air temperature in this case ($T_{surf\,snow}$) is equal to the temperature of the topmost snow layer ($T_{snow0}$). The air humidity is equal to the saturation humidity at the given temperature:

$$q_{v\,surf\,snow} = q_{v\,sat}(T_{snow\,0}), \tag{15}$$

where $q_{v\,sat}(T_{snow0})$ is the saturation air specific humidity (kg kg$^{-1}$) at snow cover surface temperature $T_{snow0}$.

$$q_{v\,sat} = \begin{cases} q_{v\,sat}^{water}(T_{snow0}), & if\ T_{snow0} \geqslant T_0 \\ q_{v\,sat}^{ice}(T_{snow0}), & if\ T_{snow0} < T_0 \end{cases}, \tag{16}$$

where $q_{v\,sat}^{water}$ and $q_{v\,sat}^{ice}$ are saturation air specific humidity (kg kg$^{-1}$) at snow cover surface temperature $T_{snow0}$ over liquid water and over ice respectively.

## 210  2.6 Air temperature and humidity over a composite soil surface

Having defined the values of air temperature and humidity over all the possible components of a composite soil surface, and knowing the fraction of each component of the surface, it is possible to define the overall surface air temperature ($T_{surf}$) and humidity ($q_{v\,surf}$).

In absence of snow cover, or in presence of a so-called shallow snow layer (see the description of snow scheme) the overall

surface air temperature is equal to the weighted mean of the temperatures of the surface components:

$$T_{surf} = T_{surf\,soil} \cdot F_{baresoil} + T_{veg}^{dry} \cdot F_{veg}^{dry} + T_{veg}^{wet} \cdot F_{veg}^{wet}, \tag{17}$$

while in case of a thick snow layer we have:

$$T_{surf} = T_{surf\,snow}. \tag{18}$$

Similar formulae hold for the surface specific humidity, in case of absence of snow cover:

$$q_{v\,surf} = q_{v\,surf\,soil} \cdot F_{baresoil} + q_{v\,veg}^{dry} \cdot F_{veg}^{dry} + q_{v\,veg}^{wet} \cdot F_{veg}^{wet}, \tag{19}$$

and in presence of snow cover (either thick or shallow):

$$q_{v\,surf} = q_{v\,surf\,snow}. \tag{20}$$

## 2.7 Entropy flux between soil surface and atmosphere

The surface incoming entropy flux is composed by the turbulent flux of entropy for dry air, the turbulent flux of entropy due to water vapour and the entropy flux due to global radiation:

$$\Phi_{S\,surf} = \Phi_{S\,rad} + \Phi_{S_{da}}^{turb} + \Phi_{S_v}^{turb}, \tag{21}$$

where:

$\Phi_{S\,surf}$ is the surface entropy flux (J K$^{-1}$ m$^{-2}$ s$^{-1}$),

$\Phi_{S\,da}{}^{turb}$ and $\Phi_{S\,v}{}^{turb}$ are the entropy fluxes originating from turbulent entropy flux of dry air and of water vapour (J K$^{-1}$ m$^{-2}$ s$^{-1}$),

$\Phi_{S\,rad}$ is the entropy flux originating from global radiation (J K$^{-1}$ m$^{-2}$ s$^{-1}$).

The flux of entropy due to the flux of water (in liquid and solid phases) from atmospheric precipitation is neglected since in the soil entropy scheme the entropy flux originating from soil moisture flux is also neglected.

The entropy fluxes are computed according to the following relations:

$$\Phi_{S\,rad} = \frac{\Phi_{rad}}{T_{surf}}, \tag{22}$$

where $\Phi_{rad}$ is the flux of global radiation (W m$^{-2}$).

$$\Phi_{S_{da}\,surf}^{turb} = K_h^{turb} \cdot \rho_{a\,surf} \cdot \frac{S_{da\,surf} - S_{da\,atm}}{z_{atm}}, \tag{23}$$

where:

$K_h{}^{turb}$ is the coefficient of heat exchange in the surface layer (m$^2$ s$^{-1}$),

$\rho_{a\,surf}$ is the air density at the surface (kg m$^{-3}$),

$z_{atm}$ is the height of the lowest atmospheric level (m).

$S_{da\,surf}$ and $S_{da\,atm}$ are specific entropy of dry air on the surface and at the lowest atmospheric level (J kg$^{-1}$ K$^{-1}$).

The entropy of dry air is defined by the relation:

$$S_{da} = q_d \cdot \left[ C_p^d ln\left(\frac{T}{T_0}\right) - R_d ln\left(\frac{P_d}{P_0}\right) \right]: \tag{24}$$

where:

$S_{da}$ is the specific entropy of dry air (J kg$^{-1}$ K$^{-1}$),

$q_d$ is the specific mass of dry air (kg kg$^{-1}$),

$T$ is the temperature (K),

$P_d$ is the partial pressure of dry air (Pa),

$T_0$=273.15 K is the reference temperature,

$P_0$=10$^5$ Pa is the reference pressure,

$C_p{}^d$=1004.6 J kg$^{-1}$ K$^{-1}$ is the specific heat capacity of dry air at constant pressure,

$R_d$=287.05 J kg$^{-1}$ K$^{-1}$ is the gas constant of dry air.

In order to define the entropy of dry air at the surface and at the lowest atmospheric level, the known values of air temperature, humidity and pressure are used together with the relations:

$$q_{da\ surf} = 1 - q_{v\ surf}, \tag{25}$$

$$q_{da\ atm} = 1 - q_{v\ atm}, \tag{26}$$

$$P_{d\ surf} = P_{surf} - e_{surf}, \tag{27}$$

$$P_{d\ atm} = P_{atm} - e_{atm}, \tag{28}$$

$q_{da\ surf}$ and $q_{da\ atm}$ are dry air specific mass at the surface and at the lowest atmospheric level (kg kg$^{-1}$),

$P_{d\ surf}$ and $P_{d\ atm}$ are partial pressure of dry air at the surface and at the lowest atmospheric level (Pa),

$e_{surf}$ and $e_{atm}$ are partial pressure of water vapour at the surface and at the lowest atmospheric level (Pa).

The entropy flux of water vapour originating from turbulent exchange in the layer between soil surface and lowest atmospheric level is defined as:

$$\Phi_{S_v}^{turb} = K_v^{turb} \cdot \rho_a \cdot \frac{S_{v\ surf} - S_{v\ atm}}{z_{atm}}, \tag{29}$$

where $S_{v\ surf}$ and $S_{v\ atm}$ are specific entropy of water vapour at the surface and at the lowest atmospheric level respectively (J kg$^{-1}$ K$^{-1}$), which, in turn, are defined by:

$$S_v = q_v \cdot \left[ C_v^d ln\left(\frac{T}{T_0}\right) - R_v ln\left(\frac{e}{e_0}\right) + \frac{L_i^v}{T_0} \right] \tag{30}$$

where:

$S_v$ is the specific entropy of water vapour(J kg$^{-1}$ K$^{-1}$),

$q_v$ is the air specific humidity (kg kg$^{-1}$),

$e$ is the partial pressure of water vapour (Pa),

$e_0$=611 Pa is the reference partial pressure of water vapour,

$C_p^v$=1869.46 J kg$^{-1}$ K$^{-1}$ is the specific heat of water vapour at constant pressure,

$R_v$=461.51 J kg$^{-1}$ K$^{-1}$ is the gas constant for water vapour,

$L_i^v$=2834170.5 J kg$^{-1}$ is the specific latent heat for ice-vapour phase transition.

The total entropy flux between atmosphere and soil surface ($\Phi_{S\ surf}$) determined in this way has to be assigned to each component of the complex soil surface in order to define the boundary condition for each type of surface. As introduced in section 2.1, the surface can either be composed of soil partially covered by low vegetation, or by snow, which, in turn, may be "thick" or "shallow". From the point of view of entropy (energy) exchange, low vegetation behaves as a "transparent" layer, i.e. it does not have an own temperature and it is part of the soil surface thus it does not have its own entropy flux. Thus, two cases may occur.

The first case occurs when snow cover is absent or shallow. In this case all the components of the flux fully impinge on the soil surface with low vegetation and snow surface does not receive any flux:

$$\Phi_{S\ surf}^{soil} = \Phi_{S\ surf},$$

$\Phi_{S\,surf}^{snow} = 0.$

The second case occurs when snow layer is present and thick. In this case all the entropy flux impinges on the snow surface and the soil surface does not receive any direct flux:

$$\Phi_{S\,surf}^{soil} = 0,$$

$$\Phi_{S\,surf}^{snow} = \Phi_{S\,surf}.$$

## 2.8 Water vapour flux between soil and atmosphere

The flux of water vapour originating by means of turbulent exchange between surface and atmosphere is defined as:

$$\Phi_v^{turb} = K_v^{turb} \cdot \rho_{a\,surf} \cdot \frac{q_{v\,surf} - q_{v\,atm}}{z_{atm}}, \tag{31}$$

where $\Phi_v^{\,turb}$ is the flux of water vapour in the atmosphere surface layer (kg m$^{-2}$ s$^{-1}$).

This summary flux has to be split between the components of the soil surface.

In the absence of snow cover, the water vapour exchange takes place between atmosphere and soil surface covered by (partly wet) vegetation. Two cases can be distinguished: in the first case the flux is positive (i.e. downwards) thus condensation (deposition) of water vapour on the surface takes place; in the second case the flux is negative (i.e. upwards) thus evaporation (sublimation) from the surface takes place.

When the flux is directed downwards, it partly impinges on the bare soil surface and partly on the vegetation, where it

contributes to the formation of dew over the leaves up to a maximum pre-specified value of water content as in the formulae:

$$\Phi_{v\,soil}^{turb} = \Phi_v^{turb} \cdot \left(1 - F_{veg}\right), \tag{32}$$

$$\Phi_{v\,veg\,dry}^{turb} = 0, \tag{33}$$

$$\Phi_{v\,veg\,wet}^{turb} = min\left\{\Phi_v^{turb} \cdot F_{veg}, \frac{q_{w\,veg}^{max} - q_{w\,veg}}{\Delta t}\right\}, \tag{34}$$

where:

$\Phi_v^{\,turb}{}_{soil}$ is the flux of water vapour in the atmospheric surface layer towards bare soil (kg m$^{-2}$ s$^{-1}$),

$\Phi_v^{\,turb}{}_{veg\,dry}$ and $\Phi_v^{\,turb}{}_{veg\,wet}$ are the fluxes of water vapour in the atmospheric surface layer towards not moistened and moistened vegetation (kg m$^{-2}$ s$^{-1}$),

$q_{w\,veg}$ and $q_{w\,veg}^{max}$ are the water content of plant leaves and its maximum value (kg m$^{-2}$) respectively,

$\Delta t$ is the model time step (s).

When the flux is directed upwards, it removes water partly from the bare soil surface, partly from the soil through plant evapotranspiration and partly through evaporation of water on the leaves. When conditions for evapotranspiration are not met, the corresponding flux is zero (see description of vegetation scheme), while evaporation from leaves obviously occurs as long as there is water available on leaf surface, as in the following formulae:

$$\Phi_{v\,veg\,dry}^{turb} = \begin{cases} \Phi_v^{turb} \cdot F_{veg}^{dry}, & if\ evapotranspiration\ possible \\ 0, & if\ evapotranspiration\ not\ possible \end{cases}, \tag{35}$$

$$\Phi_{v\,veg\,wet}^{turb} = max\left\{\Phi_v^{turb} \cdot F_{veg}^{wet}, \frac{-q_{w\,veg}}{\Delta t}\right\}, \tag{36}$$

$$\Phi_{v\,soil}^{turb} = \Phi_v^{turb} - \Phi_{v\,veg\,dry}^{turb} - \Phi_{v\,veg\,wet}^{turb}. \tag{37}$$

In the presence of snow cover, the water vapour fluxes between atmosphere and bare soil/vegetation are null and the interaction takes place only between atmosphere and snow layer:

$$\Phi_{v\,snow}^{turb} = \Phi_v^{turb}, \tag{38}$$

where $\Phi_{v\,snow}^{turb}$ is the flux of water vapour in the atmosphere surface layer towards the snow layer (kg m$^{-2}$ s$^{-1}$).

## 2.9 Atmospheric precipitation flux at the surface

Atmospheric precipitation flux over surface, divided into liquid ($\Phi_w^{liq}$) and solid ($\Phi_w^{ice}$), is provided by the atmospheric model. The distribution of these fluxes over the soil surface depends on the presence of snow over it.

In absence of snow cover, liquid precipitation contributes to the surface components, including leaves, according to the fraction
of each component; the amount of water on leaves exceeding the maximum allowable is immediately redistributed among the bare soil components:

$$\Phi_{w\,veg}^{liq} = min\left\{\Phi_w^{liq} \cdot F_{veg}, \frac{q_{w\,veg}^{max} - q_{w\,veg}}{\Delta t}\right\}, \tag{39}$$

$$\Phi_{w\,soil}^{liq} = \Phi_w^{liq} - \Phi_{w\,veg}^{liq}, \tag{40}$$

$$\Phi_{w\,snow}^{liq} = 0, \tag{41}$$

where $\Phi_w^{liq}$, $\Phi_{w\,soil}^{liq}$, $\Phi_{w\,veg}^{liq}$, $\Phi_{w\,snow}^{liq}$ are the fluxes of atmospheric precipitation in liquid phase on the whole surface, on the soil surface, on vegetation and on the snow-covered surface respectively (kg m$^{-2}$ s$^{-1}$).

In presence of snow cover, all the liquid precipitation flux is directed towards the snow layer:

$$\Phi_{w\,soil}^{liq} = \Phi_{w\,veg}^{liq} = 0, \tag{42}$$

$$\Phi_{w\,snow}^{liq} = \Phi_w^{liq}. \tag{43}$$

Conversely, solid precipitation flux is always directed towards snow layer, creating it if it does not exist:

$$\Phi_{w\,snow}^{ice} = \Phi_w^{ice}, \tag{44}$$

$$\Phi_{w\,soil}^{ice} = \Phi_{w\,veg}^{ice} = 0. \tag{45}$$

where $\Phi_w^{ice}$, $\Phi_{w\,snow}^{ice}$, $\Phi_{w\,soil}^{ice}$, $\Phi_{w\,veg}^{ice}$ are the fluxes of atmospheric precipitation in solid phase on the whole surface, on the snow-covered surface, on the snow-free soil surface and on vegetation respectively (kg m$^{-2}$ s$^{-1}$).

The surface processes scheme defines the conditions on the upper layer of the soil column in absence of snow cover or on the snow column in presence of snow. The boundary conditions are given by fluxes of entropy and water which can comprise water vapour and precipitation. The soil scheme can also define the air temperature and humidity over a composite surface.

## 3 Scheme of vegetation processes

In the vegetation scheme two processes are represented: evapotranspiration and interception of water by plant leaves. Considering evapotranspiration process, we recall that in the previous section the water vapour flux between the soil surface and the lowest atmospheric model was defined taking into account evapotranspiration of plants ($\Phi_v^{turb}{}_{veg\,dry}$, 33, 35). In this section the conditions under which evapotranspiration can take place are defined, as well as the change in soil wetness due to evapotranspiration.

Evapotranspiration is possible when the following conditions are fulfilled:

1. in each level of the plant root zone the temperature is above 0°C;
2. the air saturation specific humidity at surface temperature is higher than the actual air specific humidity at the lowest atmospheric layer, i.e. the water vapour flux can be directed upwards;
3. leaves are present, i.e. Leaf Area Index (LAI) is nonzero;

4. photosynthesis is possible, i.e. the incoming visible radiation flux is positive.

The water vapor flux due to evapotranspiration from leaf surface not covered by water ($\Phi_v^{turb}{}_{veg\,dry}$, see paragraph 2.7) removes water from the root zone of the soil and each layer of this zone loses water proportionally to its contribution to evapotranspiration flux. In paragraph 2.3 it was shown how to determine air humidity over a vegetation surface not covered by water depending on the evapotranspiration rate (6) and the scheme of vertical space discretisation in soil was presented (fig.

350    2).

Consequently, the contribution of each root-zone level to the overall water flux due to evapotranspiration becomes:

$$F_k = \frac{(z_{kh+1} - z_{kh}) \cdot F\left(q_k^{rel}\right)}{z_{root}} \tag{46}$$

and the wetness variation in each soil level due to evapotranspiration is:

$$q_k^{\Delta t} = q_k^0 + \Phi_{v\,veg\,dry}^{turb} \cdot F_k \cdot \frac{\Delta t}{\rho_w \cdot (z_{kh+1} - z_{kh})}, \tag{47}$$

where $\rho_w$ is the liquid water density (kg m$^{-3}$).

When considering interception by vegetation leaves, the water content over low-vegetation leaves is determined by the turbulent flux of water vapour between the leaves and the lowest atmospheric level ($\Phi_v^{turb}{}_{veg\,wet}$), which can lead either to evaporation (sublimation) or to condensation (deposition), as well as by atmospheric liquid precipitation flux over vegetation surface ($\Phi_w^{liq}{}_{veg}$). Thus the prognostic equation for the water deposited over leaves looks like:

$$q_{w\,veg}^{\Delta t} = max\left\{min\left[q_{w\,veg}^0 + \left(\Phi_{v\,veg\,wet}^{turb} + \Phi_{w\,veg}^{liq}\right) \cdot \Delta t, q_{w\,veg}^{max}\right], 0\right\}, \tag{48}$$

where $q_{w\,veg}^{\Delta t}$ and $q_{w\,veg}^0$ are the water content on plant leaves at the beginning and at the end of the time step (kg m$^{-2}$), while the other variables were defined at paragraphs 2.7 and 2.8.

The water intercepted by vegetation leaves can cover the leaves either partially or completely, as mentioned in paragraph 2.1, where the concept of leaf fraction covered by water ($F_{veg}^{leaf\,wet}$) was introduced. This fraction is determined by the diagnostic

relation:

$$F_{leaf\ wet}^{veg} = \left(\frac{q_{w\ veg}}{q_{w\ veg}^{max}}\right)^{2/3}, \tag{49}$$

where the exponent 2/3 is needed to obtain the cross-section ratio from the volume ratio for a spherical drop.

The vegetation scheme thus defines the variation of water content in the root zone due to evapotranspiration and to water intercepted by leaves and provides a diagnostic relation for the leaf fraction covered by water.


## 4. Scheme of water exchange processes in the soil

The main equation describing dynamics of liquid water along the soil profile is the Darcy's law:

$$\Phi_f = \frac{-\Omega}{\mu}\nabla P, \tag{50}$$

where $\Phi_f$ is the fluid flux (m s$^{-1}$), $\nabla P$ is the pressure gradient (Pa m$^{-1}$), $\mu$ is the fluid viscosity (Pa s), $\Omega$ is the section area (m$^2$).

When applied to the transport of water in soil, Darcy's law takes the form:

$$W_s\frac{\partial \Psi}{\partial t} = K\nabla^2\Psi - G, \tag{51}$$

where $\Psi$ is the hydraulic head or hydraulic potential (m), $W_s$ is the ratio of drained water volume at saturation to the total material volume (m$^3$m$^{-3}$) or maximum specific volumetric water content, $K$ is the hydraulic conductivity (m s$^{-1}$), $G$ represents the water source terms (m$^3$m$^{-3}$s$^{-1}$), $t$ is time (s).

Assuming the absence of water sources, in the hypothesis of constant $K$ and considering only the vertical coordinate, the equation can be written as:

$$W_s\frac{\partial \Psi}{\partial t} = K\frac{\partial^2 \Psi}{\partial z^2}. \tag{52}$$

In the usual "soil notation" this equation is written as:

$$q_{max}\frac{\partial \Psi}{\partial t} = \frac{\partial\left(\frac{\Phi_w}{\rho_w}\right)}{\partial z}, \tag{53}$$

where $\frac{\Phi_w}{\rho_w} = K\frac{\partial \Psi}{\partial z}$, $\Phi_w$ is the soil water flux (kg m$^{-2}$ s$^{-1}$), $q_{max}$ is the maximum specific volumetric water content (m$^3$m$^{-3}$), i.e. in the case when all the soil pores are filled with water; this parameter depends on the soil texture and changes along the profile depending on soil horizon.

The equation introduced here operates on soil hydraulic potential, while the main prognostic quantity for soil moisture is $q$, i.e. the specific volumetric moisture content (m$^3$m$^{-3}$), so it is necessary to express $\Psi$ in terms of $q$.

Using the method of Clapp and Hornberger (1978) and introducing the concept of partially frozen soil moisture, the hydraulic potential can be represented as:

$$\Psi = \Psi_g\left(\frac{q_{max}}{q\left(1-f_{ice}^{soil}\right)}\right)^b, \tag{54}$$

where $\Psi_g$ is the hydraulic potential of saturated soil (i.e. when $q=q_{max}$), $b$ is an empirical parameter named "soil exponent", both these parameters depend on soil texture and can change along the soil profile depending on soil horizon, $f_{ice}{}^{soil}$ is the fraction of frozen water with respect to total soil water. It can be noticed that formula (54) is valid only when $f_{ice}{}^{soil}<1$ since in the event of total freezing of water in the soil, the hydraulic potential tends to infinity and no moisture motion can take place. An important component of (53) is the hydraulic conductivity of soil, which depends on its physical properties and on soil water content itself. Using again the Clapp and Hornberger (1978) method and extending it to the case of partly frozen soil moisture, the dependence of hydraulic conductivity on water content takes the form:

$$K = K_g \left( \frac{q - f_{ice}^{soil} q}{q_{max} - f_{ice}^{soil} q} \right)^{2b+3}, \tag{55}$$

where $K_g$ is the hydraulic conductivity of saturated soil (when $q=q_{max}$), also depending on soil texture.

By substituting equation (54) into equation (53) under the assumption that the fraction of frozen water does not change during the process, i.e. $\partial f_{ice}^{soil}/\partial t = 0$, a prognostic equation for $q$ is obtained, describing the motion of moisture along the soil profile:

$$\frac{\partial q}{\partial t} = \frac{\frac{\partial \left( \frac{\Phi_w}{\rho_w} \right)}{\partial z}}{-\frac{q_{max}}{q} \cdot \Psi_g \cdot b \cdot \left( \frac{q_{max}}{q\left(1 - f_{ice}^{soil}\right)} \right)^b}. \tag{56}$$

In a finite difference representation, (see fig. 2), applying an explicit approximation of the moisture flux terms and of the space derivatives, equation (56) takes the form:

$$\frac{q_k^{\Delta t} - q_k^0}{\Delta t} = \frac{\frac{\Phi_{w\,kh+1}^0}{\rho_w} - \frac{\Phi_{w\,kh}^0}{\rho_w}}{z_{kh+1} - z_{kh}} \cdot \frac{1}{-\left( \frac{q_{max}}{q_k^0} \right) \cdot \Psi_g \cdot b \cdot \left( \frac{q_{max}}{q_k^0 \left(1 - f_{ice\,k}^{soil\,0}\right)} \right)^b}, \tag{57}$$

$$\frac{\Phi_{w\,kh}^0}{\rho_w} = K_{kh} \cdot \frac{\Psi_k^0 - \Psi_{k-1}^0}{z_k - z_{k-1}}, \tag{58}$$

where the upper indexes 0 and $\Delta t$ indicate the variables at the beginning and at the end of the time step respectively, while the lower indexes $k$ and $kh$ indicate the values of variables taken at full and half vertical levels respectively (fig. 2).

The values of the variables at half levels are computed as arithmetical means $x_{kh}=1/2(x_{k-1}+x_k)$.

In the case when a soil layer is completely frozen, $f_{ice}{}^{soil}=1$, the hydraulic potential tends to infinity, so the moisture flux is simply set to 0, i.e. if $f_{ice}{}^{soil}=1$ or $f_{ice\,k-1}=1$, then $\Phi^0{}_{wkh}=0$.

**5 Scheme of thermal exchange processes in the soil**

As stated in section 2, the quantity chosen to describe the thermal state of the environment in the present model is entropy. The use of this quantity allows to describe the phase changes of water in soil in a simple mathematical form, while it does not differ significantly from other thermodynamic quantities in the description of thermal exchange. In the present model two main approximations are applied in the numerical solution scheme. The first is the application of the splitting method for solving the prognostic equation for entropy, i.e. the equation for the conductive transport of entropy is solved separately from the

equation for entropy conservation in moist soil in case of phase change of soil water. The second approximation consists in neglecting the entropy due to water fluxes in the equation for the conductive transport of entropy. These approximations are applied on the basis of the experience in the numerical solution of the given problem. Due to the application of different space and time approximations, problems generated by small numerical inaccuracies (differences of big numbers) appeared, leading to an unacceptable instability and unphysical solution in particular situations. The aforementioned approximations mitigated

these instabilities.

Starting from the first part of the problem, i.e. the conductive transport of entropy in wet soil, this process is described by the diffusion equation in the form:

$$\frac{\partial S_{soil}}{\partial t} = \frac{\partial \Phi_{S\ soil}}{\partial z},$$ (59)

where $S_{soil}$ is the soil entropy (J K$^{-1}$ m$^{-3}$), $\Phi_{S\ soil}$ is the soil entropy flux (J K$^{-1}$ m$^{-2}$ s$^{-1}$).

The entropy of wet soil is a function of the specific entropy:

$$S_{soil} = \rho_{soil} S_{soil}^{spec},$$ (60)

$$S_{soil}^{spec} = C_{soil} ln\left(\frac{T}{T_0}\right),$$ (61)

where $S_{soil}^{spec}$ is the specific entropy of wet soil (J kg$^{-1}$ K$^{-1}$), $\rho_{soil}$ is the density of wet soil (kg m$^{-3}$), $C_{soil}$ is the specific heat capacity of wet soil (J kg$^{-1}$ K$^{-1}$) , $T$ is the soil temperature (K).

The conductive flux of entropy is defined as:

$$\Phi_{S\ soil} = \frac{\lambda_{soil}}{C_{soil}} \frac{\partial S_{soil}^{spec}}{\partial z},$$ (62)

where $\lambda_{soil}$ is the heat conductivity of wet soil (J s$^{-1}$ m$^{-1}$ K$^{-1}$).

From the thermodynamical point of view, wet soil includes two components: dry soil, which does not undergo phase changes, and water, which undergoes phase changes and can be represented as a mixture of water and ice (the gaseous phase of water

in soil is neglected). For this reason, the following assumptions are made in relation to the parameters of wet soil:

$$\rho_{soil} = \rho_{soil}^{dry} + q\left((1 - f_{ice}^{soil})\rho_w + f_{ice}^{soil}\rho_i\right),$$ (63)

$$C_{soil} = C_{soil}^{dry} + q\left((1 - f_{ice}^{soil})C_w + f_{ice}^{soil}C_i\right),$$ (64)

where $\rho_{soil}^{dry}$ is the density of dry soil (kg m$^{-3}$), $C_{soil}^{dry}$ is the specific heat capacity of dry soil (J kg$^{-1}$ K$^{-1}$), both quantities depending on the soil characteristics (texture) and varying along the vertical, $\rho_w$ and $\rho_i$ are density of liquid water (1000 kg m$^{-3}$) and ice (900 kg m$^{-3}$), $C_w$ and $C_i$ are specific heat capacity of liquid water (4186.8 J kg$^{-1}$ K$^{-1}$) and ice (2093.4 J kg$^{-1}$ K$^{-1}$), $q$ is

the soil specific volumetric water content (m$^3$ m$^{-3}$) and $f_{ice}^{soil}$ is the fraction of frozen water in total soil water, introduced in previous chapters.

Defining the value of the specific heat conductivity of moist soil is by itself a non-obvious problem. The main factor influencing this quantity is the moisture content of the soil. Different approaches for defining soil heat conductivity depending on its

moisture content are known in literature, for example through hydraulic potential (Pielke, 2013) or through relative water

content and heat conductivity of dry and saturated soil (Peters-Lidard et al., 1998, Best at al., 2011). In the present work a different approach is proposed. i.e. by means of relative water content and soil density:

$$\lambda_{soil} = min \left\{ min \left[ \frac{\rho_{soil}^{dry}}{1000} \cdot \sqrt{q^{rel}} + 0.3 \cdot \frac{\rho_{soil}^{dry}}{1000}, 3.0 \right] + q \cdot f_{ice}^{soil} \cdot \lambda_i, 3.0 \right\}, \tag{65}$$

where $q^{rel}$ is the soil relative water content, as in (13) and $\lambda_i$ is the heat conductivity of ice (2.0 J s$^{-1}$ m$^{-1}$ K$^{-1}$).

The proposed definition and the proposed values for the coefficients were formulated during the numerical experiments and verification of air temperature shown in section 9. The definition of this quantity significantly influences near-surface temperatures, especially daily minimum and maximum values and amplitude of diurnal cycle, in the cases of stable boundary layer. The experiments showed that the given formula is suitable for different types of soil encountered in the territories of Europe and Western Asia.

Considering now the second part of the problem, i.e. the conservation of entropy of soil moisture in case of phase change, the quantity which has to be conserved is the sum of liquid water and ice entropy:

$$S_{soil}^{water} = \rho_w q \left(1 - f_{ice}^{soil}\right) \left( C_w ln \frac{T}{T_0} + \frac{L_i^w}{T_0} \right) + \rho_i q f_{ice}^{soil} C_i ln \frac{T}{T_0}, \tag{66}$$

where $S_{soil}^{water}$ is the entropy of soil water (J K$^{-1}$ m$^{-3}$), $L_i^w$ is the specific latent heat of ice-water phase change (333560.5 J kg$^{-1}$).

The solution of the basic prognostic soil entropy equation (59) is split into two steps: the conductive transport of the entropy (eq. 69 see below) and the phase change of the soil water. In the approximation of the conductive entropy transport the water phase changes are not considered, i.e. the fraction of ice in total soil water ($f_{ice}^{soil}$) is assumed to be known and the only unknown is the temperature. In the equation describing the phase changes of soil water (66) two unknowns are present, temperature and fraction of ice in soil water, so that, in order to solve this equation an additional equation relating the two quantities has to be

added. This equation is introduced on the basis of the hypotheses that at temperatures over 0 ºC the fraction of ice is equal to zero, while at temperatures below a certain threshold (here -30 ºC is assumed) the water in liquid phase cannot exist thus the fraction is equal to one. Between these two threshold values the fraction of ice grows monotonically with decreasing temperature and the shape of the growth is assumed to be a hyperbolic tangent:

$$f_{ice}^{soil} = -tanh[(T - T_0) \cdot a \cdot f_b], \tag{67}$$

where the empirical coefficient $a = \frac{-4}{-30}$ defines the thermodynamic regime and does not depend on the soil characteristics, while the coefficient $f_b$ depends on the soil characteristics and can assume values in the interval $1 \le f_b \le 2$:

$$f_b = 2 - \left\{ \frac{min[max(b, 4), 12] - 4}{8} \right\}, \tag{68}$$

where $b$ is the soil exponent already introduced in previous sections (about water exchange processes), the higher the value of $b$, the smoother the growth of the ice fraction with decreasing temperature.

The numerical solution of the split problem is now considered: the discretisation of the vertical coordinate is the one shown in section 2 (fig.2), while a time-explicit approximation of fluxes and of their derivatives, is used; the equation for conductive transport of entropy (59) thus becomes, in finite-difference form:

$$\frac{\rho_{soil}^k C_{soil}^k ln\frac{T_k^*}{T_0} - \rho_{soil}^k C_{soil}^k ln\frac{T_k^0}{T_0}}{\Delta t} = \frac{\frac{\lambda_{soil}^{kh+1} C_{soil}^{k+1} ln\frac{T_{k+1}^0}{T_0} - C_{soil}^k ln\frac{T_k^0}{T_0}}{C_{soil}^{kh+1}} \frac{\lambda_{soil}^{kh} C_{soil}^k ln\frac{T_k^0}{T_0} - C_{soil}^{k-1} ln\frac{T_{k-1}^0}{T_0}}{C_{soil}^{kh}}}{z_{kh+1} - z_{kh}},$$ 

(69)

where indices $k$ and $kh$ indicate values on vertical full and half levels respectively, the upper indices $^o$ and $^*$ indicate values of temperature before and after the solution of the conductive transport equation respectively. The values of the physical parameters on half levels are computed as the arithmetical mean of the values on the full levels. In order to compute the density and the heat capacity of wet soil, the value of soil ice fraction computed at the beginning of the step is used.

The solution of (69) allows to compute the temperature T* taking into account the contribution of conductive heat flux but without taking into account any possible phase change.

After solving the first part of the split problem, the obtained value of temperature T* allows to compute the entropy of soil water, which is being considered as its final value at the end of the time step:

$$S_{soil\,k}^{\Delta t} = \left\{\rho_w q_k \left(1 - f_{ice\,k}^{soil\,0}\right)\left(C_w ln\frac{T_k^*}{T_0} + \frac{L_i^w}{T_0}\right) + \rho_i q_k f_{ice\,k}^{soil\,0} C_i ln\frac{T_k^*}{T_0}\right\} \cdot (z_{kh+1} - z_{kh}),$$

(70)

where $q_k$ and $f_{ice\,k}^{soil\,0}$ are the total moisture content and soil ice fraction on the level $k$, before taking into account the phase changes.

The value of soil water entropy obtained is then used for computing the temperature and ice fraction at the end of the time step, i.e. after considering the possible phase changes, by solving the equation system:

$$\begin{cases} \left[\rho_w q_k \left(1 - f_{ice\,k}^{soil\,\Delta t}(T_k^{\Delta t})\right)\left(C_w ln\frac{T_k^{\Delta t}}{T_0} + \frac{L_i^w}{T_0}\right) + \rho_i q_k f_{ice\,k}^{soil\,\Delta t}(T_k^{\Delta t}) C_i ln\frac{T_k^{\Delta t}}{T_0}\right] \cdot (z_{kh+1} - z_{kh}) = S_{soil\,k}^{\Delta t}, \\ f_{ice}^{soil\,\Delta t} = -tanh[(T_k^{\Delta t} - T_0) \cdot a \cdot f_b] \end{cases}$$

(71)

where $T_k^{\Delta t}$ and $f_{ice\,k}^{soil\,\Delta t}$ are the values of temperature and ice fraction on level $k$ at the end of the time step. The system (71) is solved by successive iterations, which is an effective method in this case since functions (66) and (68) are smooth and monotonous.

It should be noted that, in the presence of a thin layer of snow above the soil surface, for which it is not convenient, from the point of view of numerical precision, to solve a separate equation for conductive transport and phase change, the entropy of soil moisture on the upper soil layer is increased by the value of the entropy of the thin snow layer (see next section for more details). The resulting temperature value is valid both for the soil surface and for the snow layer.

## 6 Snow scheme

The processes of formation, transformation and melting of snow above the soil surface are very important as they are associated with water phase changes, i.e. with a powerful energy source or sink, and with an important thermal insulating layer between atmosphere and soil. In this work an original multilayer scheme for the evolution of snow cover is proposed.

As it was shown in section 2, from the point of view of thermodynamic processes, the snow layer may either cover the entire surface or be completely absent. However, a concept of minimal snow thickness is introduced, above which the snow can be

considered as a separate layer in terms of heat transport and phase changes. If the snow layer thickness is smaller than this minimum value, the snow is considered as an additional component of the soil surface (section 5). At the same time, when considering water balance, i.e. processes related to atmospheric precipitation and water vapour condensation and sublimation, the snow layer thickness can be arbitrarily small, i.e. there is no minimum layer thickness.

The snow layer can be approximated as a porous ice mass which can contain water in the liquid phase, formed either by melting of the mass itself or because of liquid precipitation incoming. This liquid water, as soon as it appears, flows in the deepest layers of snow or in the soil. On the snow surface, sublimation from the solid ice phase takes place.

The proposed model uses the snow mass per unit area ($kg/m^2$) as a vertical coordinate in the snow layer instead of the more common geometric length, so the term "layer thickness" here refers to the amount of snow mass associated with a layer rather

than its geometrical thickness. The vertical discretisation includes full and half levels, the topmost full level has index zero, the level index increases with depth and each half level is situated above the full level with the same index (fig. 3). With the use of this vertical coordinate each layer, except the topmost one, has the same, constant standard thickness. An increase or decrease of total snow mass first changes the thickness of the top layer. If this thickness reaches or exceeds the standard thickness or becomes smaller than a minimum value, a layer is added or removed respectively. In these cases the values of

snow temperature and melted water content are recomputed considering the newly appeared or disappeared level so that, throughout the snow column, the total snow entropy, the liquid water content and other diagnostic characteristics such as snow age and density are conserved. However, the amount of vertical snow levels cannot exceed a given value. When the snow cover thickness is such that this amount of vertical levels is not enough, the standard layer thickness is increased (doubled) for that point and all the prognostic and diagnostic quantities are recomputed on the new set of levels with conservation of the

vertical integral values. The opposite happens when, in case of snow mass reduction, the number of levels becomes too small, in that case the standard thickness is reduced (halved) up to the initial standard thickness. In this way the numerical scheme allows to represent a snow cover of arbitrary thickness, follow its thickening or thinning, while keeping the number of layers between given limits.

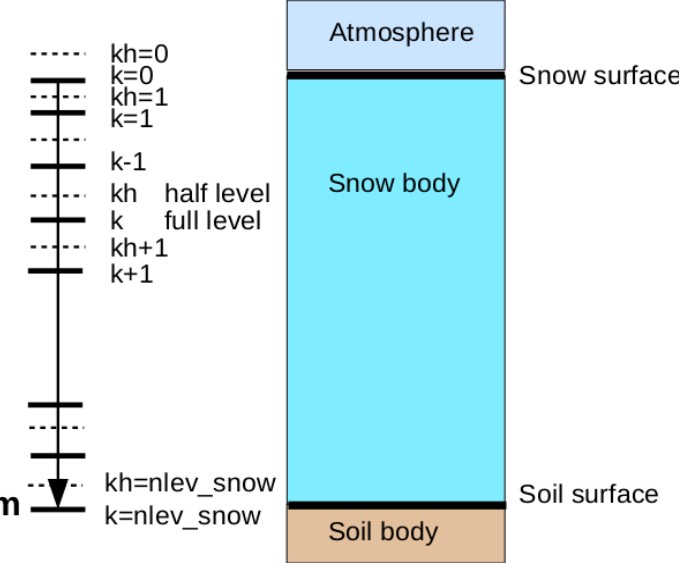

**Figure 3. Scheme of the finite difference representation of the space coordinate in the snow layer.**

## 6.1 Dynamics and balance of snow mass

The dynamics of the snow layer mass is determined by the variations of the two components of the snow layer: solid and liquid. The variations of the liquid component take place in presence of liquid precipitation falling and condensation of water
vapour on the top snow level and in case of snow melting in the whole layer; in this case liquid water flows into the lower layers or in the soil. The general balance of snow mass is determined by the sum of the water fluxes (in all the phases) at the top and bottom layers.

The water mass flux at the top layer is described in section 2 (see equations 38 ($\Phi_v^{turb}{}_{snow}$), 41 and 43 ($\Phi_w^{liq}{}_{snow}$), 44 ($\Phi_w^{ice}{}_{snow}$)). At the bottom layer, the water mass flux is determined by the liquid water flux from the layer above. The process of water
draining along the snow profile is described in the following way: all the liquid water that at the beginning of the time step is found at level k, at the end of the step is found at level k+1, this hypothesis is acceptable since empirical data show that even a very small liquid water draining speed is anyway higher than the values resulting from this hypothesis. The thickness of the snow layers is in the order of centimetre and the time step is in the order of a minute, thus all the water can drain through a layer of any reasonable thickness a few seconds even with very low hydraulic conductivity valueof the order of $10^{-4}$ m/s. The
liquid water flux at a half snow layer is thus:

$$\Phi_{m\,kh} = \frac{m_{k-1} \cdot (1 - f_{ice\,k-1}^{snow})}{\Delta t},$$ (72)

where $\Phi_{m\ kh}$ is the flux of liquid water at level $kh$ (kg m-2 s-1), $m_{k-1}$ is the specific (total) snow mass at level $k-1$ (kg m-2), $f_{ice}^{snow}{}_{k-1}$ is the fraction of ice phase in the total snow mass at level $k-1$.

The water flux at the lowest snow half level is the water flux at the soil surface (see section 3).

## 6.2 Processes of heat conduction and water phase transition in the snow

As in the soil scheme, the main equation describing the thermodynamic state of the snow is the equation of entropy transport and conservation (see section 5). In the case of snow, for solving the entropy prognostic equation the splitting method is applied: first, the conductive entropy transport term is solved, then the entropy conservation in case of phase transition in the snow layer is solved.

Considering the conductive transport, in mass coordinates, the equation has the following aspect:

$$\frac{\partial S_{snow}^{spec}}{\partial t} = \frac{\partial \Phi_{S\ snow}}{\partial m}, \tag{73}$$

where $S_{snow}^{spec}$ is the specific entropy of snow (J K-1 kg-1) and $\Phi_{S\ snow}$ is the snow entropy flux (J K-1 m-2 s-1).

By analogy with soil entropy (see equations 60, 61) the total specific entropy of the snow layer, including the solid and liquid phases of water, is defined as:

$$S_{snow}^{spec} = C_{snow}^{\Sigma} ln \frac{T}{T_0}, \tag{74}$$

where $C_{snow}^{\Sigma}$ is the total specific heat capacity of snow including ice and liquid water (J kg-1 K-1), which can be rewritten, by making use of the concept of fraction of solid phase with respect to total mass, in the following way:

$$C_{snow}^{\Sigma} = f_{ice}^{snow} C_i + (1 - f_{ice}^{snow}) C_w. \tag{75}$$

The flux of conductive entropy transport is defined as:

$$\Phi_{S\ snow} = \frac{\lambda_{snow}}{C_{snow}^{\Sigma}} \cdot \rho_{snow}^{\Sigma} \cdot \frac{\partial S_{snow}^{spec}}{\partial m}, \tag{76}$$

where $\lambda_{snow}$ is the specific heat conductivity of snow (J s-1 m-1 K-1) and $\rho_{snow}^{\Sigma}$ is the total density of snow including ice and liquid water (kg m-3).

In this equation the density is not the density of the porous medium but it is a virtual density of the thermodynamically active medium, excluding the pore volume, defined as:

$$\rho_{snow}^{\Sigma} = f_{ice}^{snow} \rho_i + (1 - f_{ice}^{snow}) \rho_w. \tag{77}$$

At the same time, a density from the point of view of the porous medium is introduced for the snow, in order to define the characteristics of heat conductivity. This density is indicated with the symbol $\rho_{snow}$. This quantity is a diagnostic parameter defined at each snow layer, depending on the thickness of the snow layer, on the snow age at each layer and on the total period during which the snow at every layer was subject to melting/freezing processes. The following method for determining diagnostic snow density is proposed in the present model:

$$\rho_{snow} = min[f_2 \cdot f_3, \rho_{firn}]$$

$$f_2 = max\left\{min\left[\left(\frac{\tau_{snow}^{melt}+30}{395}\right)^{0.3}, 1\right] \cdot \rho_{firn}, f_1\right\}$$

$$f_1 = max\left\{min\left[\left(\frac{\tau_{snow}}{365}\right)^{0.3}, 1\right] \cdot \rho_{snow}^{old}, \rho_{snow}^{fresh}\right\},$$

$$f_3 = min\left[1 + 0.5\left(\frac{m}{100}\right)^{0.5}, 1.5\right]$$

$$(78)$$

where $\tau_{snow}$ and $\tau_{snow}^{melt}$ are the total snow age and the total period during which snow was subject to melting (days), $\rho_{firn}$, $\rho_{snow}^{fresh}$ and $\rho_{snow}^{old}$ are density of firn, of fresh snow and of old snow (kg m-3), $m$ is the snow mass in the current layer

according to the vertical coordinate used (kg m-2). When density is determined, a limitation is applied, according to which the density variation cannot exceed 10% per day.

The heat conductivity of snow was defined following the study of (Jin et al., 1999):

$$\lambda_{snow} = 2.45 \cdot 10^{-6} \cdot \rho_{snow}^2. \tag{79}$$

When equation (73) is solved, the snow temperature including effect of heat conduction is determined, while the solid fraction

does not change and is considered known.

We consider now the solution of the second part of the problem, i.e. the conservation of snow entropy in case of phase transition. The following quantity, equal to the entropy of a particular snow layer including liquid and solid phases, should be conserved:

$$S_{snow} = \Delta m \cdot \left\{f_{ice}^{snow} C_i ln\frac{T}{T_0} + (1 - f_{ice}^{snow})\left(C_w ln\frac{T}{T_0} + \frac{L_i^w}{T_0}\right)\right\}, \tag{80}$$

where $S_{snow}$ is the entropy of snow (J K-1 m-2) and $\Delta m$ is the specific mass of a snow layer (kg m-2).

In order to numerically solve equation (73) a discretisation of the vertical coordinate as shown in fig. 3 is used, together with a time-explicit method of approximation of fluxes and of their derivatives. The following finite-difference prognostic equation is thus obtained:

$$\frac{c_{snow\,k}^{\Sigma} ln\frac{T_k^*}{T_0} - c_{snow\,k}^{\Sigma} ln\frac{T_k^0}{T_0}}{\Delta t} = \rho_{snow\,k}^{\Sigma} \cdot \frac{\frac{\lambda_{snow}^{kh+1}}{c_{snow\,kh+1}^{\Sigma}}\frac{c_{snow\,k+1}^{\Sigma} ln\frac{T_{k+1}^0}{T_0} - c_{snow\,k}^{\Sigma} ln\frac{T_k^0}{T_0}}{m_{k+1}-m_k} - \frac{\lambda_{snow}^{kh}}{c_{snow\,kh}^{\Sigma}}\frac{c_{snow\,k}^{\Sigma} ln\frac{T_k^0}{T_0} - c_{snow\,k-1}^{\Sigma} ln\frac{T_{k-1}^0}{T_0}}{m_k-m_{k-1}}}{m_{kh+1}-m_{kh}}, \tag{81}$$

where the indexes $k$ and $k_h$ indicate the values on the full and half vertical levels respectively, the upper indexes 0 and * indicate the values of temperature variables before and after the solution of the conductive heat transport equation respectively. The values of the physical parameters at half levels are computed as the arithmetic mean of the values at the surrounding full levels. In order to compute the overall virtual density (density used in thermodynamic context) and heat capacity of snow, the value of solid fraction of snow layer at the beginning of the time step is used.

Solution of (81) allows to compute temperature $T^*$ after taking into account the conductive heat transfer but without considering the phase transition.

After having solved the first part of the split problem, which provided the value of temperature $T^*$, it is possible to define the value of snow entropy, which is considered the definitive value at the end of the time step. For determining entropy, the finite different discretisation of (80) is used:

$$S_{snow\,k}^{\Delta t} = \left\{ f_{ice\,k}^{snow\,0} C_i ln\frac{T_k^*}{T_0} + \left(1 - f_{ice\,k}^{soil\,0}\right)\left(C_w ln\frac{T_k^*}{T_0} + \frac{L_i^W}{T_0}\right)\right\} \cdot (m_{kh+1} - m_{kh}), \qquad (82)$$

where $f_{ice\,k}^{snow\,0}$ is the fraction of solid phase at level $k$ before taking into account phase transitions.

Equation (82) includes two unknowns: temperature and fraction of solid phase. Unlike the case for soil moisture, it is assumed that in the snow layer the presence of water in liquid phase ($f_{ice}^{snow}<1$) is possible only at 0°C ($T=T_0$). This assumption simplifies the solution: either the initial temperature is $T_0$, then equation (82) has a single unknown, i.e. fraction of solid phase, or the initial temperature is below zero, then the only unknown is temperature which has to be below or equal to 0 °C. The temperature and fraction of solid phase determined in this way are considered final at the end of the time step.

As remarked in section 5, the solution of the thermodynamic state of snow layer is performed only when the amount of snow exceeds a given threshold in order to avoid numerical problems. If the snow specific mass is below the given threshold, its thermodynamic state is described by the solution of the entropy equation for the top soil level, whose entropy is augmented by the value of entropy for snow.

Solution of equation (82) allows to diagnose the length of the time interval during which the snow is exposed to melting. The value of this interval is required for computing snow density.

In conclusion, the scheme of snow-layer processes defines the overall specific snow mass, the distribution of this mass in the vertical levels, and, at each level, the temperature, the fraction of solid phase, the snow age, the length of melting time-interval and the snow density.

## 7 Verification of «Pochva» scheme in NWP model Bolam in hindcast regime

The "Pochva" scheme, described above, was developed for use in numerical atmospheric models as a parameterisation of heat and water exchange processes at the surface. However, it can be also applied in a column variant using observational data on energy and water fluxes at the surface. Such application allows for studies of soil physical parameters, as well as for testing the scheme itself. The results of such testing using observations from the Coordinated Energy and Water Cycle Observation ProjectBaltic Sea Experiment (BALTEX) are presented in the appendix A.

The "Pochva" scheme is available as free and open source software and it has been coded in a way that makes it easily adaptable to any atmospheric model. The input data required are atmospheric variables at the lowest level, together with fluxes of precipitation, visible and infrared radiation, heat and humidity (or variables allowing to compute these fluxes). Besides these variables, quantities defining the physical characteristics of soil and vegetation are required. Soil characteristics are allowed to vary along the vertical direction as well. In "Pochva" the bottom boundary condition for temperature and soil moisture can be specified through a "climatological level", with a horizontally-varying depth, depending on the local climatological and

hydrological conditions, at which the values of temperature and moisture content are considered constant. The description of the methods used to define the physical parameters of soil and vegetation and the method to define the climatological level

may be the subject of a future publication. Here we only note that the spatial variability of soil physical parameters has been defined on the basis of the FAO dataset (FAO Unesco, 1997), the vegetation types and corresponding physical parameters have been defined using the GLC2000 dataset (Joint Research Centre, 2003), while, for defining the climatological levels, the analysis of air temperature at 2 m above surface for the period 1979-2014 from ECMWF ERA Interim dataset together with the FAO soil type dataset has been used.

In order to test the implementation, the "Pochva" scheme has been included in the NWP model "Bolam" (Buzzi et al., 1994, Buzzi et al., 1998) and in its global variant "Globo" (Malguzzi et al., 2011). Bolam is a hydrostatic NWP model on a limited area. A numerical experiment in hindcast regime has been set up with Bolam, the experiment consisted in a continuous integration of the model on a long period using objective analysis data as boundary conditions during all the period. As initial and boundary conditions, data from the ECMWF IFS model have been used from the ECMWF operational archive. The model

domain covered most of European territory. The time extent of the experiment covered the period June 2013-November 2015, with the first six months being used to allow the soil layer to reach the thermodynamic equilibrium with the climatological bottom boundary conditions. Consequently, these six months were excluded from the subsequent analysis of the results. In this way the effective period includes two full years, from the beginning of December 2013 to the end of November 2015. The length of two years was chosen in order to exclude the presence of interannual oscillations and trends in the simulations.

In order to verify the results, data from standard meteorological observations from WMO GTS network have been used, retrieved from ECMWF archive. The main purpose of the numerical experiment was to evaluate the contribution of the "Pochva" scheme to the numerical modelling results, thus the variables used in the verification process were air temperature and dew-point temperature at 2 m above surface.

The experiment showed that the main scores, such as mean error (bias) and root mean square error (RMSE), of near-surface

temperature and humidity, stratified on monthly and seasonal intervals do not vary significantly among the two simulated years. This suggests that there is no significant trend due to error accumulation in the simulation. In the figures below, scores averaged on seasonal intervals based on the two simulated years are shown.

Figures 4 and 5 show seasonal averages of temperature bias and RMSE over observation points. As it can be seen from fig. 4, the bias obtained is relatively low, mostly between -1ºC and +1ºC. A higher error is noticed in the cold seasons (winter,

autumn), when Central Europe experiences bias values between -1ºC and -3ºC (up to -5ºC in mountainous areas) while at the east of Ural range the bias sign is opposite (+1ºC +2ºC). In the warm seasons, mainly in summer, a bias up to +5ºC is noticed in the desert or semi-arid areas of Eastern Mediterranean.

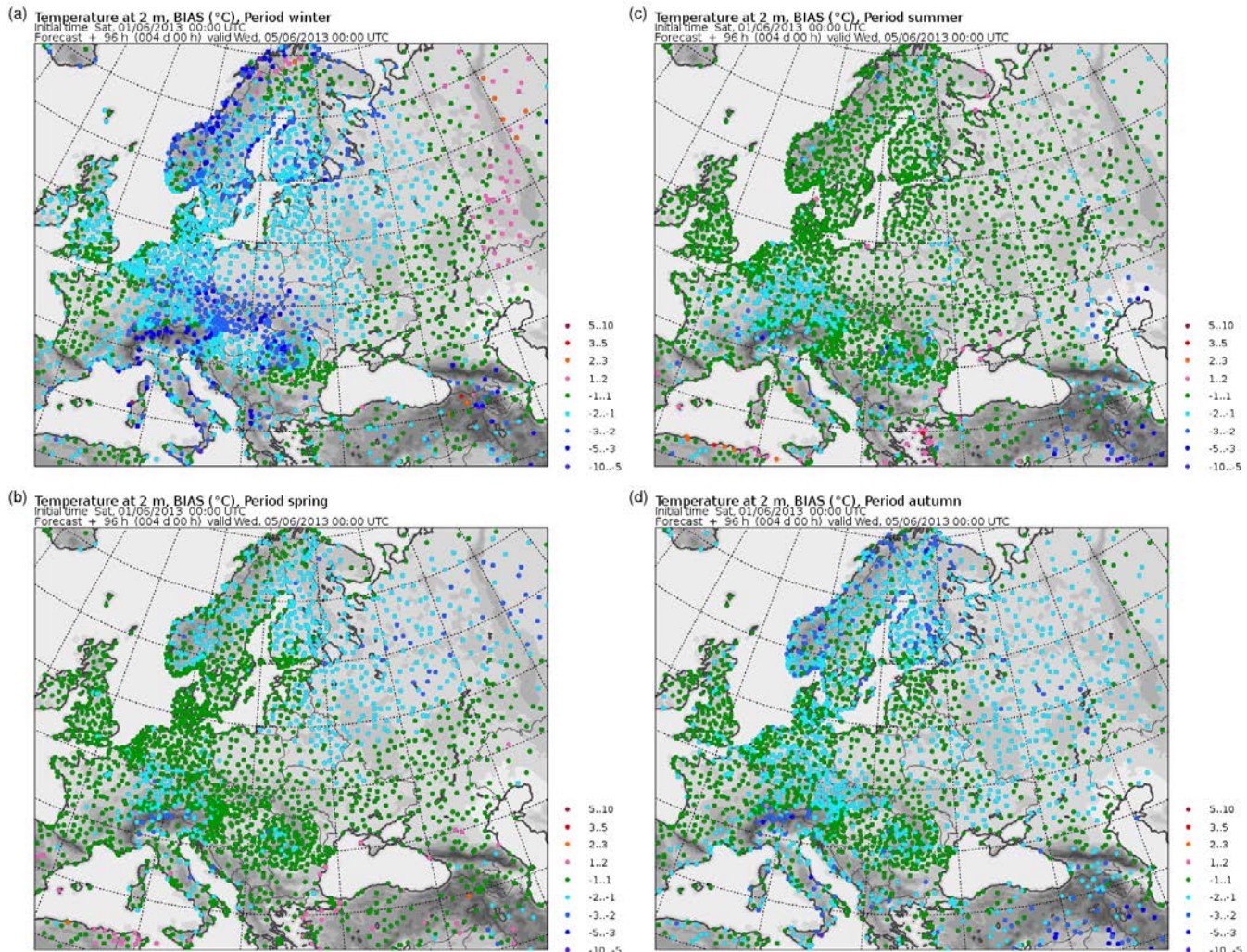

**Figure 4. Seasonally averaged bias of temperature at 2 m above surface at observation points: (a) winter, (b) spring, (c) summer, (d) autumn.**

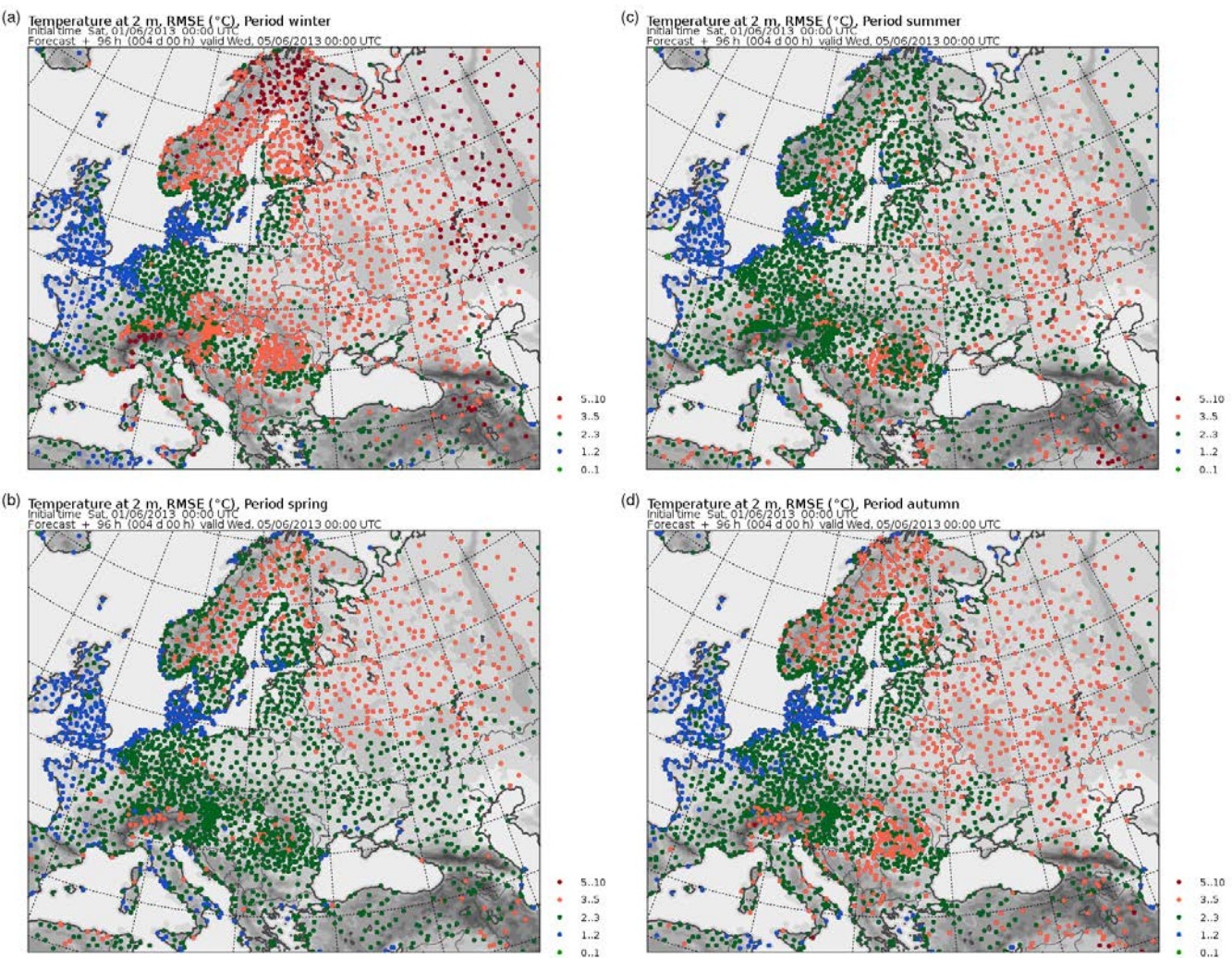

**Figure. 5. Seasonally averaged RMSE of temperature at 2 m above surface at observation points: (a) winter, (b) spring, (c) summer, (d) autumn.**


The RMSE shown in fig. 5 also indicates that, on the background of an error with a magnitude of 2-3°C, the error is higher in the colder seasons especially in mountain and continental areas, up to 5°C. In general, coastal areas show lower RMSE, with values lower than 2°C, growing up to 3-5°C with increasing distance from the sea.

It can be concluded that the overall near-surface atmospheric thermal regime is simulated with enough accuracy. The errors in
the cold periods in continental and mountain areas can be attributed to a poor simulation of the snow cover, which strongly influences the near-surface temperature, while high errors in desert areas during the warm periods can either be due to an error in the soil surface temperature or to an inaccurate representation of surface turbulent exchange in cases of dry thermal instability.

Fig. 6 shows the seasonal dew-point temperature bias on observation points. In the cold seasons the systematic error is in the interval -1ºC +2ºC, while in the warm seasons the dew-point temperature (and thus air humidity) is overestimated in the continental areas, with a bias growing up to +3ºC +5ºC with increasing distance from the sea. Similar conclusions may be drawn from the analysis of seasonal RMSE, presented in fig. 7. In the coastal areas all year round and anywhere in the colder seasons, RMSE has low or moderate values, 1-3ºC, while in the continental areas in the warm seasons it may grow up to 5ºC and more. It can thus be noticed that most of the RMSE is explained by the bias. The overestimation of air humidity in the continental areas during warm seasons is difficult to explain. The absence of a corresponding systematic error for temperature suggests that it is not due to an overestimation of evaporation, since, if that were the case, temperature would have been underestimated. It can be assumed that this is due to inaccuracy in the definition of water vapour fluxes or of the humidity profile in the surface layer in the warm season, i.e. in cases of neutral or unstable stratification.

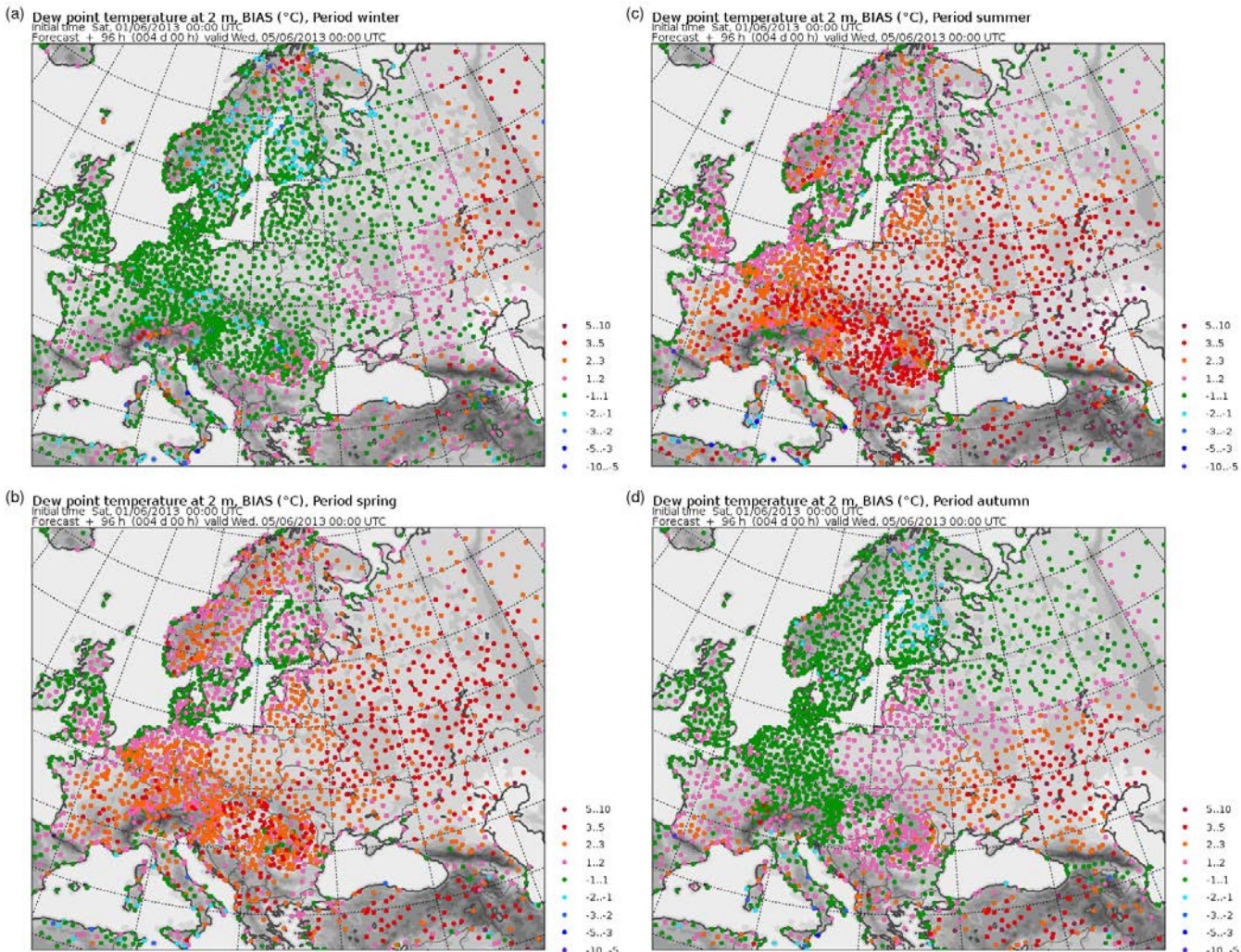

**Figure 6. Seasonally averaged bias of dew-point temperature at 2 m above surface at observation points: (a) winter, (b) spring, (c) summer, (d) autumn**

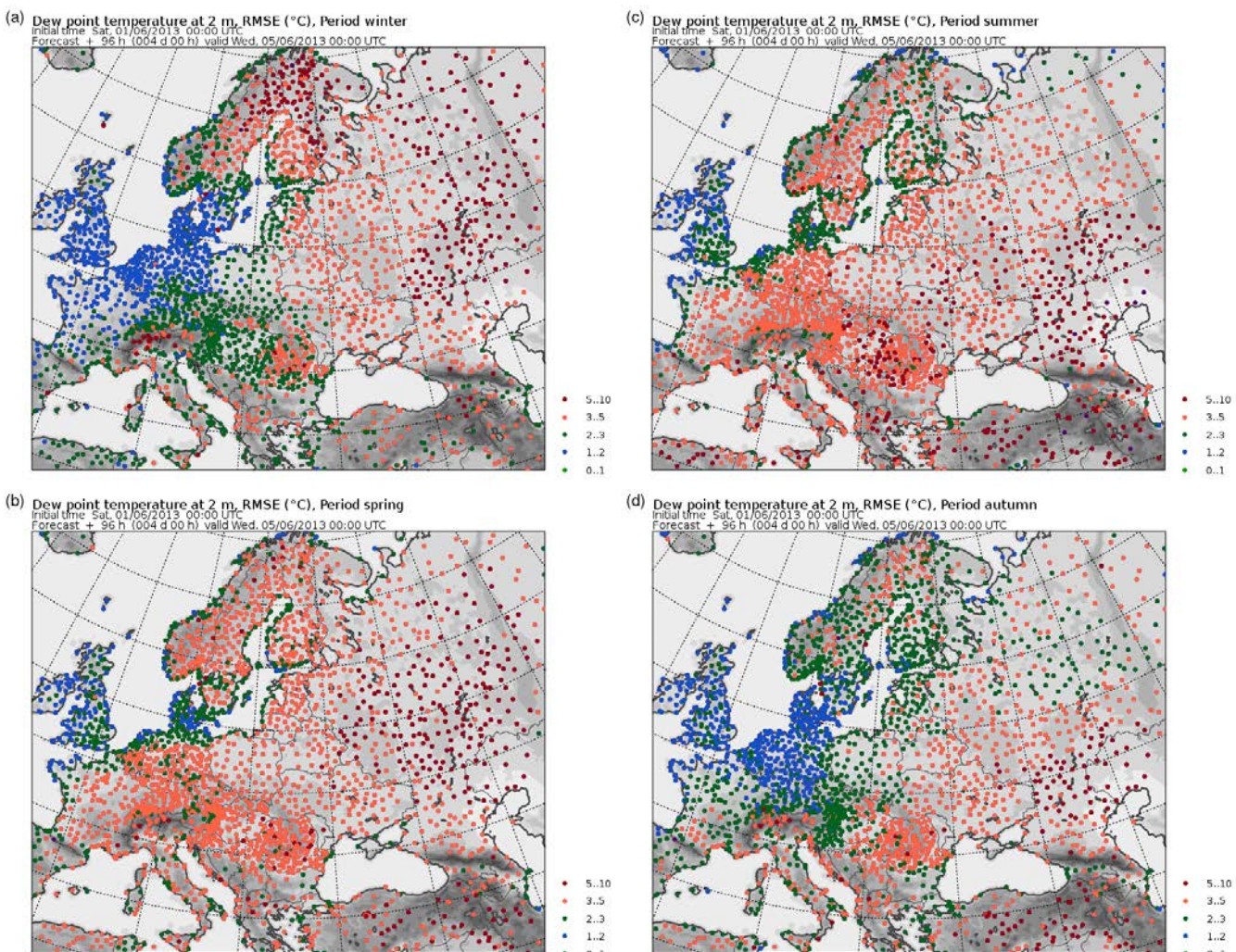

**Figure 7. Seasonally averaged RMSE of dew-point temperature at 2 above surface at observation points: (a) winter, (b) spring, (c) summer, (d) autumn.**


For a more detailed examination of the errors in near-surface air temperature and humidity from the point of view of the representation of daily cycle, the simulated and observed values for these variables at all the observation times are shown here, averaged on each season and on specific geographical areas. Since the model domain covers areas with completely different meteorological and climatological characteristics, the geographical averaging has been carried out considering climatologically

uniform areas. For this purpose, a dataset of the Köppen-Geiger climate classification has been used (Rubel and Kottek, 2010). Most of the observation points falls into six climatic areas according to the Köppen-Geiger classification, namely Bsk (cold steppe), Cfa (humid subtropical), Cfb (temperate oceanic), Csa (hot-summer Mediterranean), Dfb (warm-summer humid

continental), Dfc (subarctic). Furthermore, a separate treatment was adopted for points on mountainous areas, defined as points whose altitude exceeds 1000m above mean sea level. Fig. 8 shows the distribution of observation points by climatic zone.


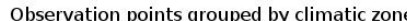
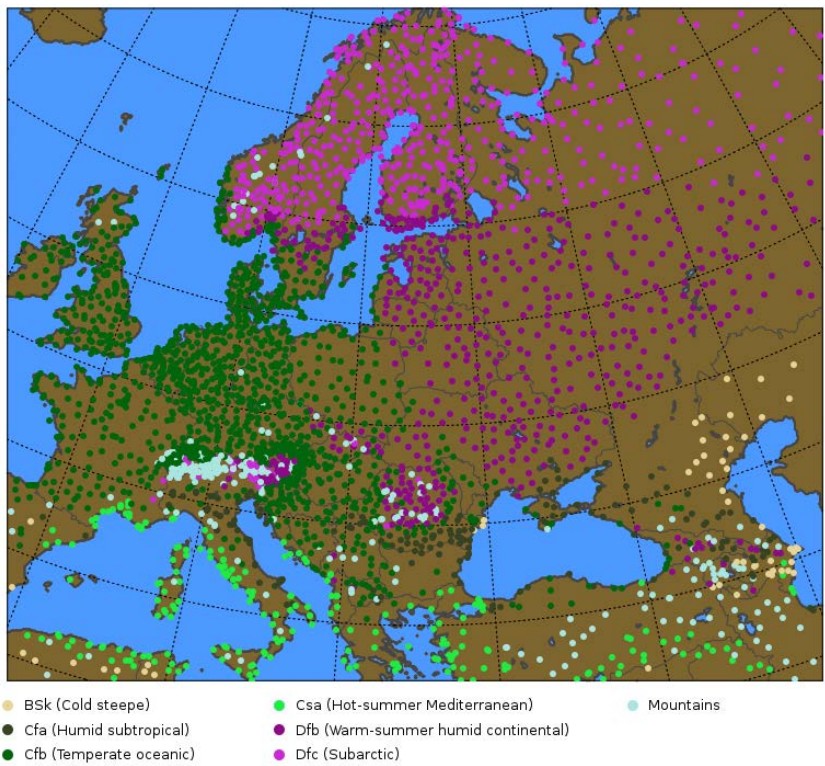

**Figure 8. Distribution of observation points by climatic zone according to Köppen-Geiger classification.**

For all the observation points falling in each of the seven main climatic zones, bias and RMSE of temperature and dew-point temperature at 2 m above surface have been computed at all the times of the day at which observations are available, namely

0, 3, 6, 9, 12, 15, 18, and 21 UTC. Below, the error obtained in the representation of the daily cycle in each climatic zone will be examined.

In winter period (fig. 9 and 10), the error does not depend on the time of the day in all the climatic zones except in the mountainous areas, where the error is higher during the day. In most of the climatic zones, the error is mostly due to bias, which is quite low, -1.3ºC -1ºC, while RMSE is in the range 1-4ºC. Error is lower in the oceanic climate areas and it increases

in areas with a continental climate. The zone with subarctic climate stands out from this picture, since it is characterised by low bias (-1.5ºC) and high RMSE (4.5-5ºC). At the same time, the mountainous areas, characterised by a similar cold climate, show good scores. This suggests that there are some problems with surface albedo definition in subarctic area and further studies are needed.

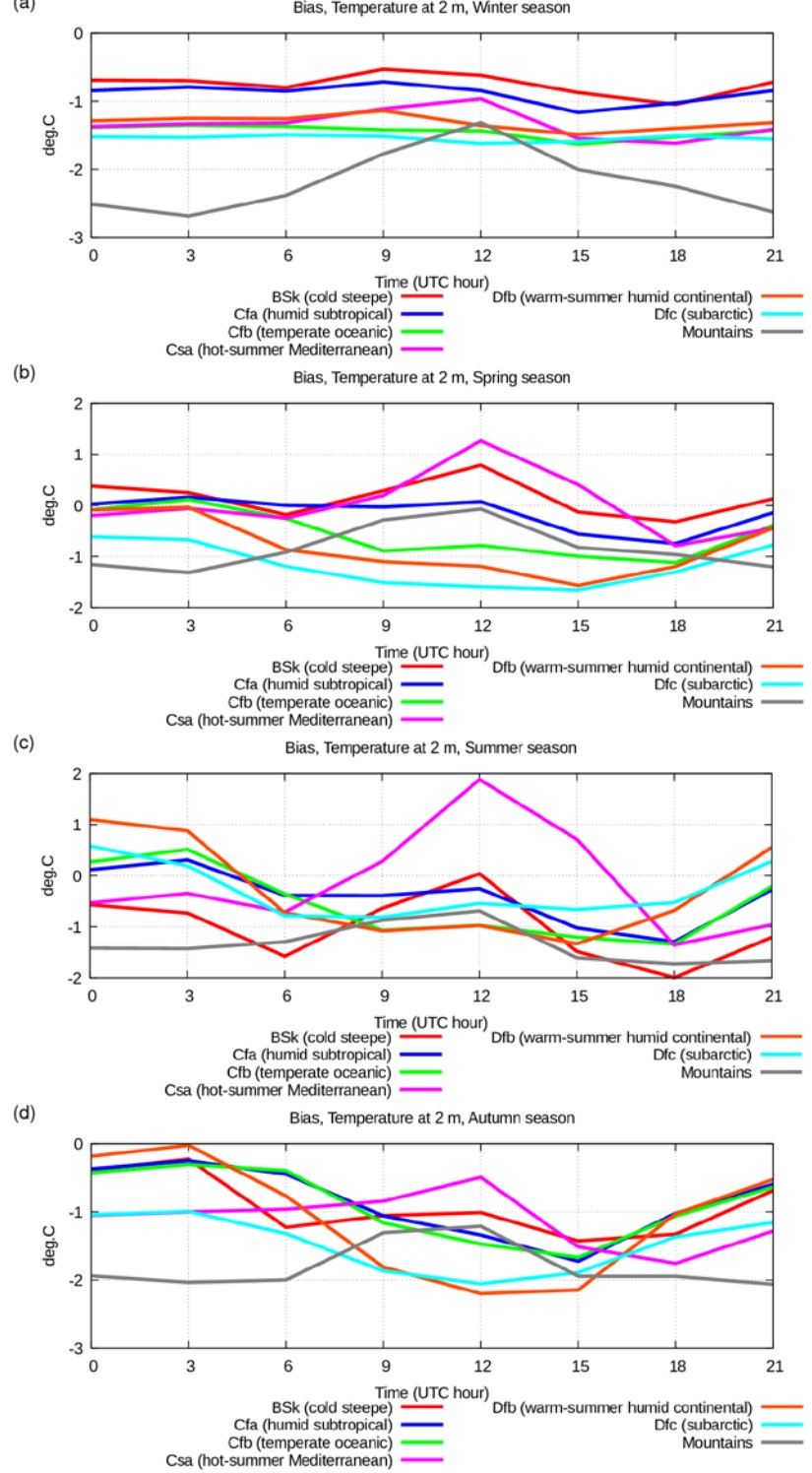

**Figure 9. Diurnal cycle of bias of simulated temperature at 2 m above surface for various seasons: (a) winter, (b) spring, (c) summer, (d) autumn, in various climatic zones: cold steppe (Bsk) red line, humid subtropical (Cfa) blue line, temperate oceanic Cfb green line, hot-summer Mediterranean (Csa) violet line, warm-summer humid continental (Dfb) orange line, subarctic (Dfa) azure line, mountains grey line.**

765

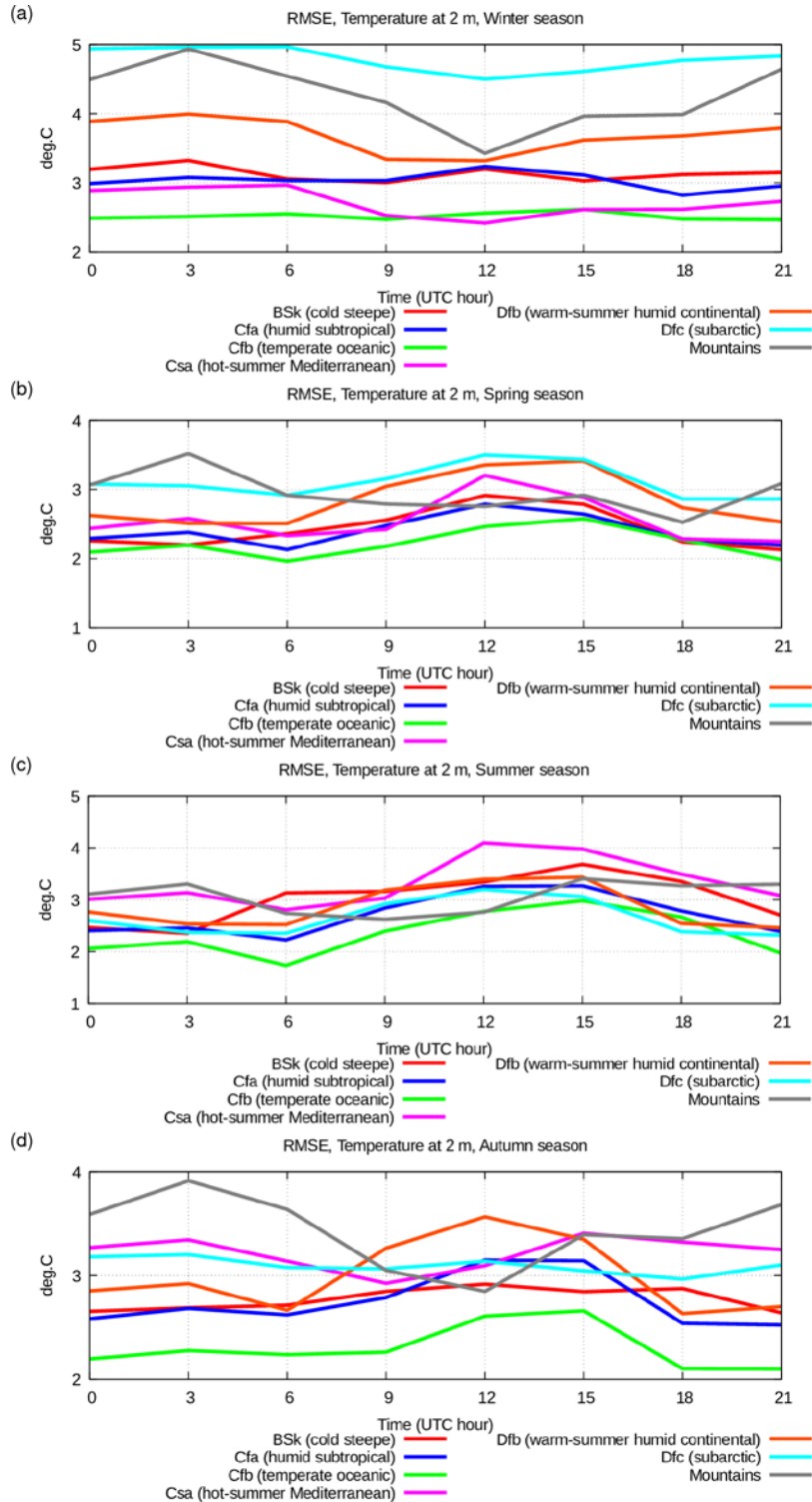

(a) RMSE, Temperature at 2 m, Winter season

(b) RMSE, Temperature at 2 m, Spring season

(c) RMSE, Temperature at 2 m, Summer season

(d) RMSE, Temperature at 2 m, Autumn season

**Figure 10. Same as figure 9 but for RMSE of simulated air temperature at 2 m above surface.**

During the summer period (fig. 9 and 10), all the climatic zones are characterised by a relatively low RMSE (2-4ºC), mostly explained by bias, which oscillates between -1.5ºC and +1ºC. Bias is higher during daytime in "hot summer Mediterranean" climate. This could be possibly due to some problems in the representation of the surface sensible heat flux in the presence of an unstable surface layer, or in the surface latent heat flux, or due to the insufficient heat capacity of the higher soil layer which, in turn, could be due to too low values of soil moisture. In areas with mountain and subarctic climate, the score is significantly better in summer than in winter. This is possibly due to deficiencies in the representation of snow cover or of the radiative characteristics of the snow cover itself.

In the spring and autumn periods, (fig. 9 and 10, panels b and d) the scores for 2-m temperature have intermediate values between those for summer and winter. The overall scores are good, the bias ranges between -1.5ºC and +1.5ºC, while RMSE is around 2-4ºC. In spring the error characteristics are closer to the summer ones, while in autumn they are closer to the winter ones. It is probably the case that the processes of formation and melting of the snow layer in the areas with a stable winter snow cover (warm-summer humid continental, subarctic and mountains) are simulated more or less correctly, since no increase of the error is observed in these transitional seasons.

Concerning the scores for humidity in terms of 2 m dew-point temperature, figures 11 and 12 show the daily cycle of the bias and RMSE for this variable.

In the winter period (fig. 11 and 12, panel a) all the climatic zones are characterised by a low systematic error (0ºC +2ºC) and a significant RMSE (2-5ºC) with a very weak daily cycle. In the same way as for temperature, the scores strongly depend on the climatic zone: better scores (RMSE up to 3ºC) are found in the area with oceanic climate (temperate oceanic, hot-summer Mediterranean, humid subtropical), while the worst scores (RMSE higher than 4ºC) are found in zones with cold continental climate (warm-summer humid continental, subarctic, mountains). This may be attributed to deficiencies in the definition of latent heat flux or air humidity profile over snow layer, i.e. in cases of stable surface layer.

In the summer period, on the other hand, most of the RMSE (2.5-8.5ºC) is explained by systematic error (0ºC +6ºC) which is always positive, i.e. the air humidity is systematically overestimated. In general, the minimum of daily error occurs at daytime, while it is maximum in the evening and night. This may be due to a suboptimal tuning of the turbulent exchange parameterisation in neutral and stable boundary layer conditions. In the areas characterised by oceanic climate or in cold climate areas (temperate oceanic, subarctic) the errors are lower, while in dry areas (cold steppe) they are higher. At the time of day when errors are higher, the overestimation of air humidity is accompanied by underestimation of air temperature. This may be an evidence of deficiencies in the computation of latent heat flux (evaporation).

During the spring and autumn seasons (fig. 11 and 12, panels b, d) the dew point temperature scores, similarly to the case of temperature, exhibit intermediate values between those found in winter and summer seasons. In general, the air humidity is almost always overestimated, more so in the warm season than in the cold season.

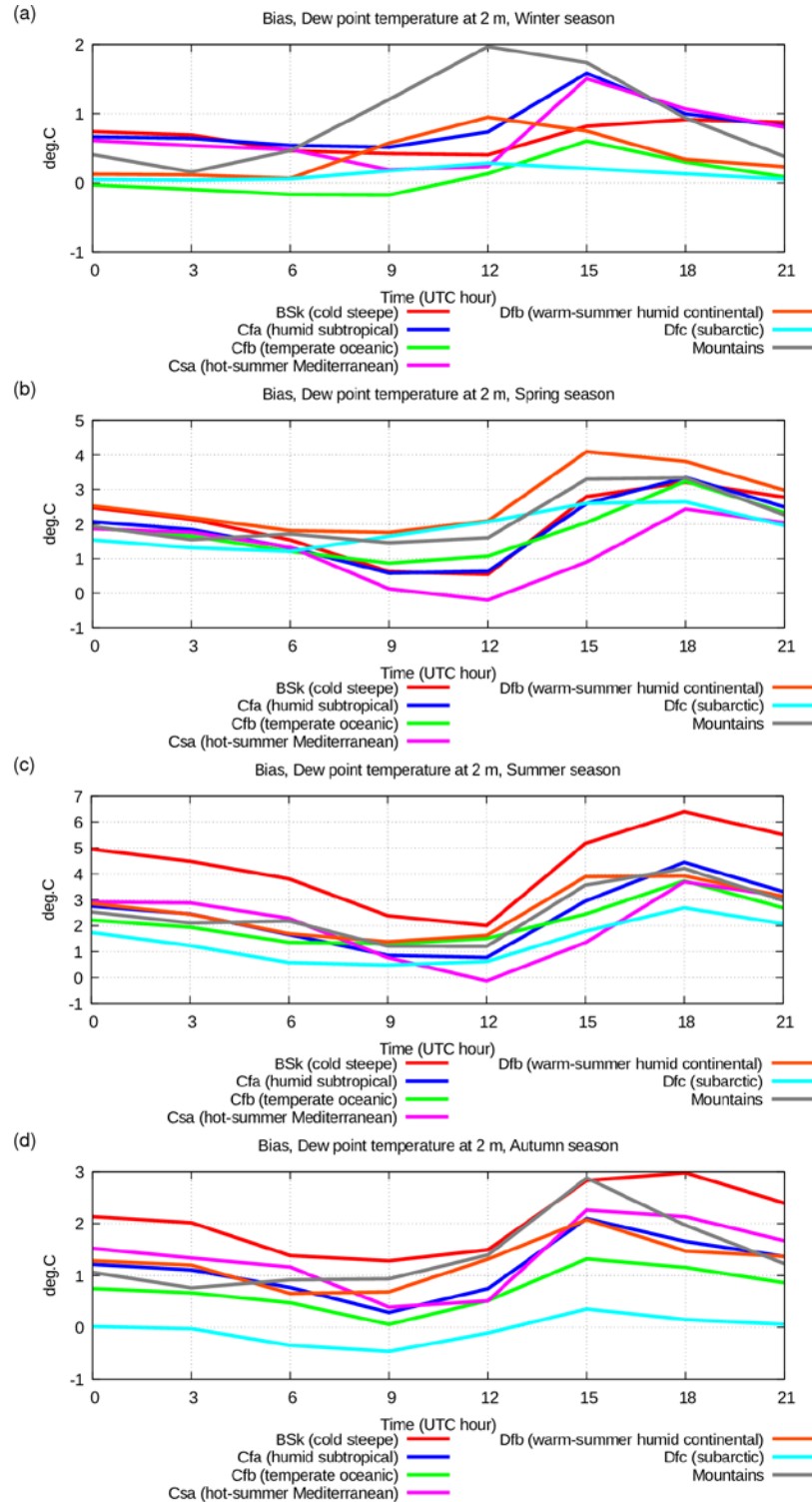

**Figure 11.** Diurnal cycle of bias of simulated dew-point temperature at 2 m above surface for various seasons: (a) winter, (b) spring, (c) summer, (d) autumn, in various climatic zones: cold steppe (Bsk) red line, humid subtropical (Cfa) blue line, temperate oceanic Cfb green line, hot-summer Mediterranean (Csa) violet line, warm-summer humid continental (Dfb) orange line, subarctic (Dfa) azure line, mountains grey line.


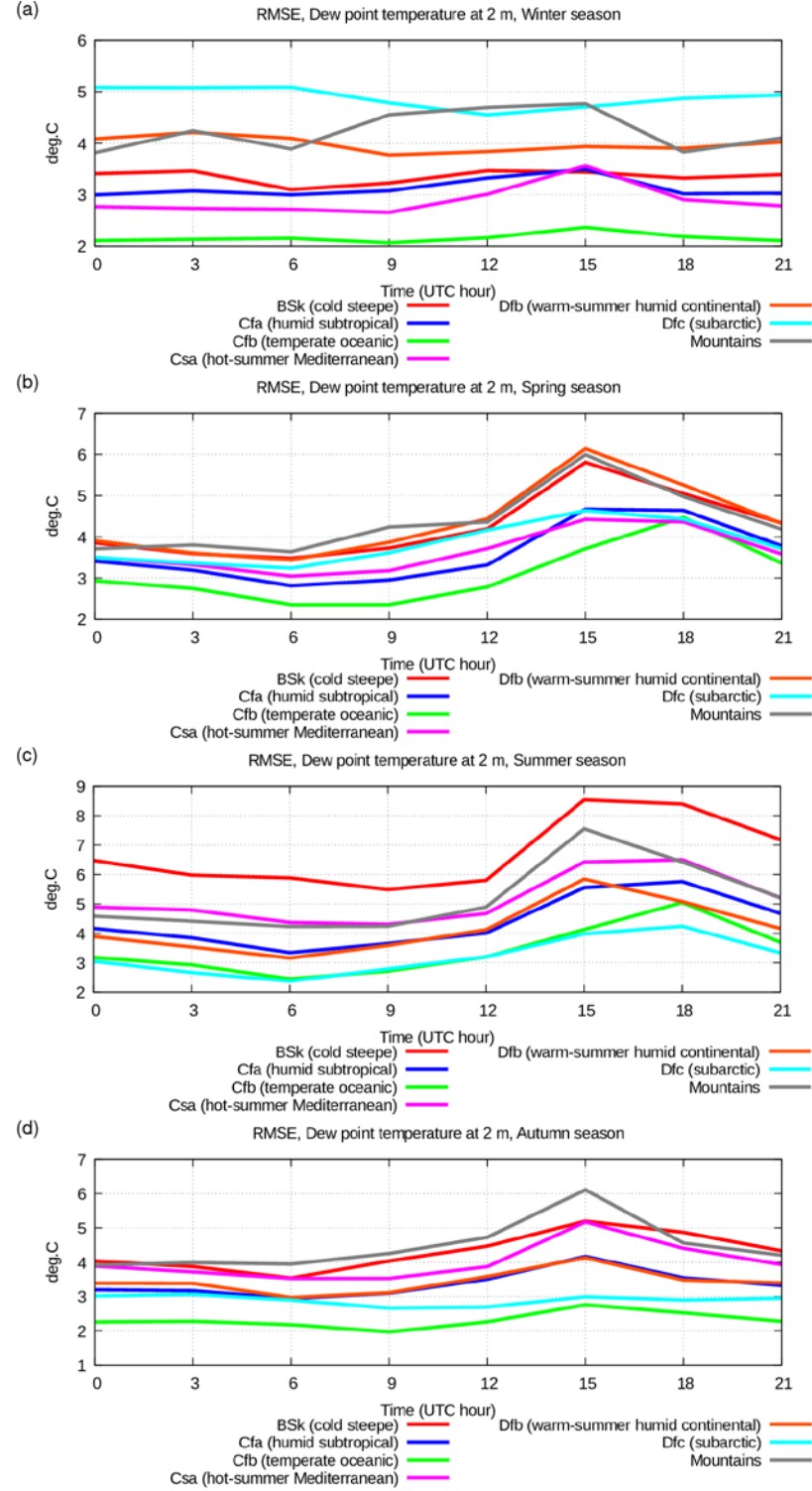

**Figure 12. Same as figure 11 but for RMSE of simulated dew-point temperature at 2 m above surface.**


The scores presented here show no evidence of systematic deficiencies in the near-surface temperature, except during periods characterised by a stable snow cover, when temperature is systematically underestimated, or during summer periods in dry climatic zones when daytime temperature is overestimated. At the same time, near-surface air humidity is systematically overestimated, especially in summer time, and, in particular, in dry climatic areas in the evening and at night.

As a general conclusion the results shown by the numerical experiments with the "Pochva" scheme are definitely convincing.

## 8 Summary

The model of hydro-thermal processes in vegetated soil and snow cover presented in this work is characterised by a special attention to the soil-water phase transition, and by novel approaches to define some soil and snow physical parameters. The proposed model has been validated by a solid verification presented in this work, consisting in a two year hindcast experiment.

The presented model may be useful for modelling research over polar or cold climate zones, in permafrost evolution studies, in studies and forecast of snow cover, in studies of the surface layer in stable conditions over cold surfaces. The model is realized using an effective and stable numerical algorithm with a clear, intuitive interface that allows a simple coupling to an atmospheric model or to observations of energy and water fluxes in the air surface layer and at the surface.

This land model has been included in the CNR-ISAC NWP models: Globo (hydrostatic approximation, global domain), Bolam,

described above, and Moloch (non-hydrostatic, high space resolution at limited area). The indicated NWP models with "Pochva" are used from 2018 up to present day for the routine operational weather prediction in CNR-ISAC for Civil Protection Department of Italy. Forecast products are available at https://www.isac.cnr.it/dinamica/projects/forecasts/ and results of forecast verification are available at https://www.isac.cnr.it/dinamica/projects/forecast_verif.

## Code availability

The model code in stand-alone version for column test simulations is available from the project website https://gitlab.com/oxana-meteo/pochva-stand-alone under the GNU GPL licence. The version of the model used to produce the results shownin this paper (v1.1) is archived on Zenodo (Drofa, 2024), this package includes the instructions for use and input/output data format description in readme.txt file.

The package of CNR-ISAC models, including "Pochva" scheme is open source code freely available at https://gitlab.com/isac-

meteo/globo-bolam-moloch.

## Competing interest

The authors declare that they have no conflict of interest.

**Acknowledgments**

The author is grateful to Francesco Tampieri (CNR-ISAC, Italy) and Davide Cesari (ARPAE-SIMC, Italy) for valuable help in article review and useful suggestions.

The author would like to thank Dmitriy Pressman (Hydrometeorological Research Center of the Russian Federation) for the knowledge that she has gained in the field of physical process modelling in soil and in NWP model development.

The author wishes to express her gratitude to the main CNR-ISAC NWP models developers Piero Malguzzi and Andrea Buzzi

(CNR-ISAC, Italy), with whom the author has co-operated for more than 20 years in model development; the author has gained invaluable experience when working with these colleagues.

The author is grateful to Fabrizio Roccato (CNR-ISAC, Italy) for his support with the computing system involved in the numerical simulations.

The author would like to thank the Italian National Computing Centre "CINECA" for providing the computational resources

on which the simulations, presented in this paper, were performed.

**Appendix A**

A test of "Pochva" scheme in its column version has been performed using observational data of Coordinated Energy and

Water Cycle Observation Project (CEOP) Baltic Sea Experiment (BALTEX). The observed data have been used as forcing parameters for "Pochva" simulation, for determining the physical parameters of soil profile and for defining the initial condition. The observed data have also been used for the verification of simulation results. The observations were carried out at Falkenberg, a site of the German Weather Service Meteorological Observatory. This site is located in grassland fields, in a heterogeneous rural landscape typical for Central Europe. The observational data are freely available upon individual request.

To apply the soil scheme offline, the values of the soil physical parameters and the soil temperature and water content profiles must be defined at the beginning of the simulation and top and bottom conditions for the prognostic variables must be defined for each time step during the simulation. A time step of 1 minute was used in the simulation. Values of soil physical parameters in Pochva simulation have been defined following the values measured at the observation point and presented in table A1 taken from dataset documentation (Beyrich, BALTEX: Lindenberg Soil Temperature and Soil Moisture Data Set., 2011). The

initial condition for prognostic variables, soil temperature and soil water content can be defined using the measurements at the soil levels. The bottom boundary conditions for prognostic variables can be defined using observation data at the lowest measurement level interpolated in time. The top boundary condition for prognostic variables, surface temperature, surface air humidity, surface soil water content, can be defined by heat and water mass fluxes at the surface: radiation flux (global net and

visible incident), sensible and latent heat flux, precipitation flux and water vapour flux. The indicated surface fluxes have to be defined using the measurement data at the surface and in the air surface layer.

The observation dataset contains measurement for the following variables: soil temperature at 0.05, 0.10, 0.15, 0.20, 0.30, 0.45, 0.50, 0.60, 0.90, 1.00, 1.20, 1.50 m depth, specific volumetric soil water content at 0.08, 0.15, 0.30, 0.45, 0.60, 0.90 m depth, pressure and temperature at the surface, air temperature and air humidity at 2.4 m above the surface, wind at 10 m above the surface, pressure, air temperature, air humidity and wind at 40 and 80 m above the surface, downward and upward short-wave and long-wave solar radiation at the surface, precipitation at the surface, sensible and latent heat flux 2.4 m above the surface. These data are presented in (Beyrich, BALTEX: Lindenberg Soil Temperature and Soil Moisture Data Set., 2011), (Beyrich, BALTEX: Lindenberg Meteorological Tower Data Set., 2011).

To avoid vertical interpolation of simulated data when comparing with observations, the Pochva simulation uses the same soil levels where soil temperature and water content measurements are available, namely: 0.02, 0.05, 0.08, 0.10, 0.15, 0.20, 0.30, 0.45, 0.50, 0.60, 0.90, 1.00, 1.20, 1.50 m depth. The bottom condition for soil temperature is defined at 1.5m depth using the bottom temperature measurement level, whereas the bottom condition for soil water content is defined at 0.9 m depth using the bottom measurement level for this parameter. Since the lower boundaries for the prognostic variables in these simulations are not deep, a few days of simulation may be sufficient to reach the numerical thermodynamic equilibrium regime.

The indicated measurement list contains all the parameters required for defining the top soil boundary condition. These parameters are also named "atmospheric forcing parameters". They are the following: sensible and latent heat flux at 2.4 m above the surface, downward and upward short-wave and long-wave radiation at the surface, precipitation at the surface (water vapour flux at the surface cat be estimated using observed latent heat flux).

| layer no. | horizon | upper boundary [cm] | lower boundary [cm] | clay / poor clay [M%] | sand [M%] | dry density [g/cm$^3$] | pore volume [%] | field capacity [V%] [*] | wilting point [V%] | hydraulic conductivity [cm/d] | soil heat capacity [*10$^6$ J/(K*m$^3$)] |
|---|---|---|---|---|---|---|---|---|---|---|---|
| Lindenberg – Falkenberg station | | | | | | | | | | | |
| 1 | Ap | 0 | 30 | 26 | 74 | 1.6 | 37 | 16 | 4 | 110 | 1.32 |
| 2 | Al | 30 | 60 | 26 | 74 | 1.7 | 36 | 18 | 3 | 80 | |
| 3 | Bt | 60 | 120 | 40 | 60 | 1.7 | 34 | 24 | 11 | 20 | |

**Table A1. Physical parameters of the soil at the Lindenberg - Falkenberg observation station from (Beyrich, BALTEX: Lindenberg Soil Temperature and Soil Moisture Data Set., 2011).**

The observation datasets contain the data for the period 01/01/2002-31/12/2009. All observed data have time resolution equal to 30 minutes. The author has developed a program that reads data from the database for various parameters, identifies the times at which data for all the atmospheric forcing parameters are available, and produces a list of the times found. Another program finds periods without gaps in the obtained list. A third program searches in the database the measurements for all required forcing parameters within the periods of continuous measurements, interpolates these data in time with the step specified for the model simulation (1 minute) and prepares all necessary data for the Pochva simulation.

Unfortunately, this search revealed very frequent gaps in the measurements of surface heat fluxes. A search for periods with gaps of no more than two hours for all the required forcing parameters resulted in 90 periods, the longest of which had a length of 4 days. However, if heat fluxes are excluded from the search of continuous measurement periods, then a very long period of 913 days is found. Therefore, the author decided to perform two types of simulations: one exclusively with the "Pochva" model using all atmospheric forcing parameters from measurements, the other with the "Pochva" model coupled with a model of air surface layer processes, which simulates surface heat fluxes using measurements of wind, pressure, air temperature, air humidity at a height of 40 m and at the surface.

The author selected two periods for the first type of Pochva simulations, one in summer and one in winter. The initial conditions have been defined using the observational data at the date and time of the simulation start. The simulation results and their comparison with the observational data are presented below.

The summer verification period is 01/07/2003 23:00 – 04/07/2003 04:30 (53.5 hours). Input data on energy fluxes at the surface and water mass fluxes are shown in figure A1.

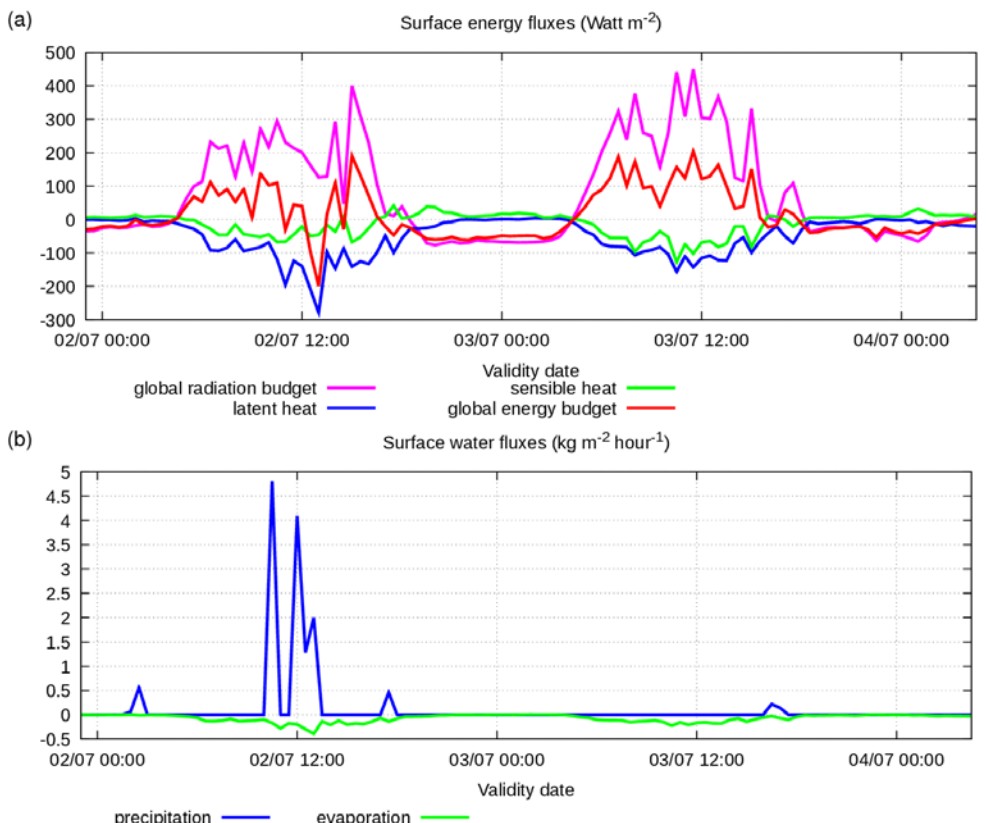

**Figure A1. Observation data (BALTEX) in Falkenberg 2-3 July 2003, parameters at the surface: (a) solar radiation and heat fluxes (W m⁻²); (b) precipitation and evaporation rate (kg m⁻² hour⁻¹).**


These July days are characterised by intensive net radiation flux accompanied by moderate sensible and latent heat fluxes, latent heat flux is significant only after light rain in the first day.

The simulated soil temperature and moisture data are shown in figures A2 and A3 together with observed data at the same depth levels.


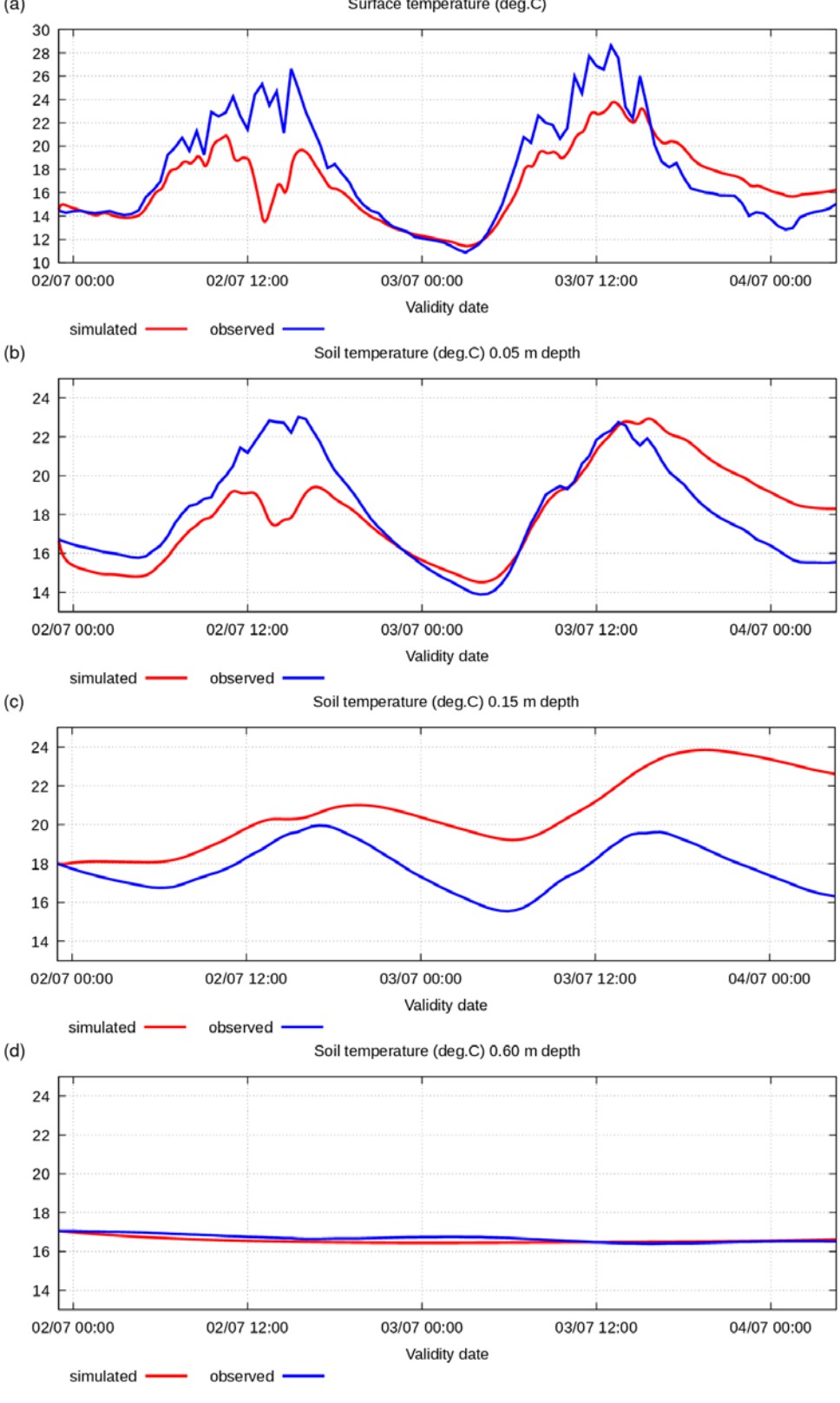

**Figure A2. Simulated and observed 2-3 July 2003 soil temperature (degree C) at the surface and at various depths: (a) surface; (b) -0.05 m; (c) -0.15 m; (d) -0.60 m.**

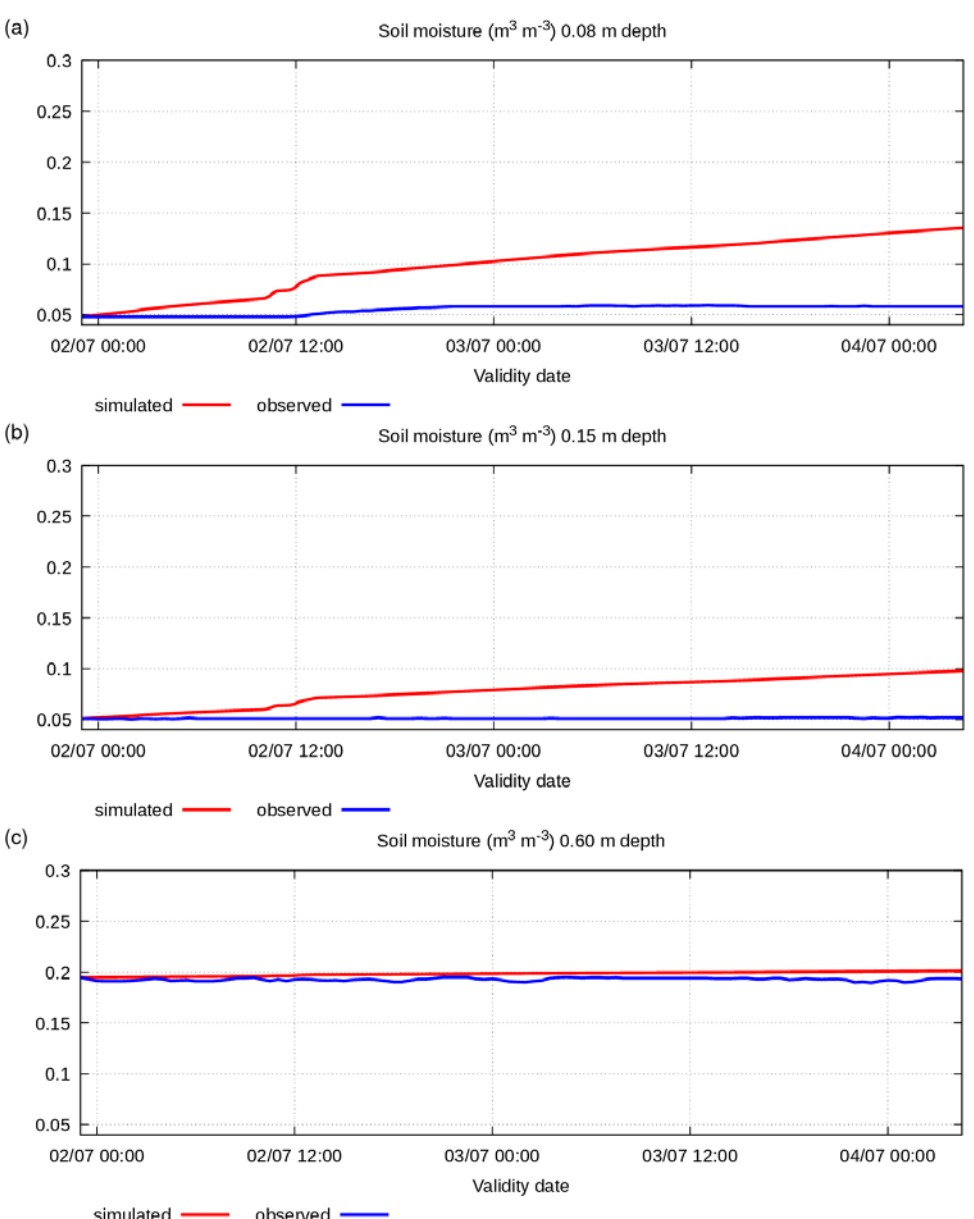


**Figure A3. Simulated and observed 2-3 July 2003 soil specific volumetric water content (m3 m-3) at various depths: (a) -0.08 m; (b) -0.15 m; (c) -0.60 m.**

Figures A2 and A3 show that the simulated soil parameters at the different levels are out of numerical equilibrium. Despite this fact, the simulated parameter values are close to observed values. The soil temperature simulated at various levels shows a correct daily cycle especially at the surface and at the top soil levels, which are important for an atmospheric model parameterisation. Moreover, the observation data show some odd characteristics: a light rain was observed on 02/07 at 12:30 (fig. A1(b)) accompanied by a sharp decrease in solar incoming radiation flux and an increase in outgoing latent heat flux, thus determining a strongly negative energy budget at the surface. This phenomenon is simulated by Pochva, the simulated surface and soil top temperature evidently drop at this time, while, on the other hand, the observed surface and soil top temperature do not reflect this phenomenon. Furthermore, the observed soil moisture at different levels remains unchanged despite the presence of significant vertical gradient of soil moisture, rain and evaporation in the observation data.

The winter verification period is 08/01/2005 17:30 – 12/01/2005 12:30 (91 hours). Input data on energy fluxes at the surface and water mass fluxes are shown in figure A4.

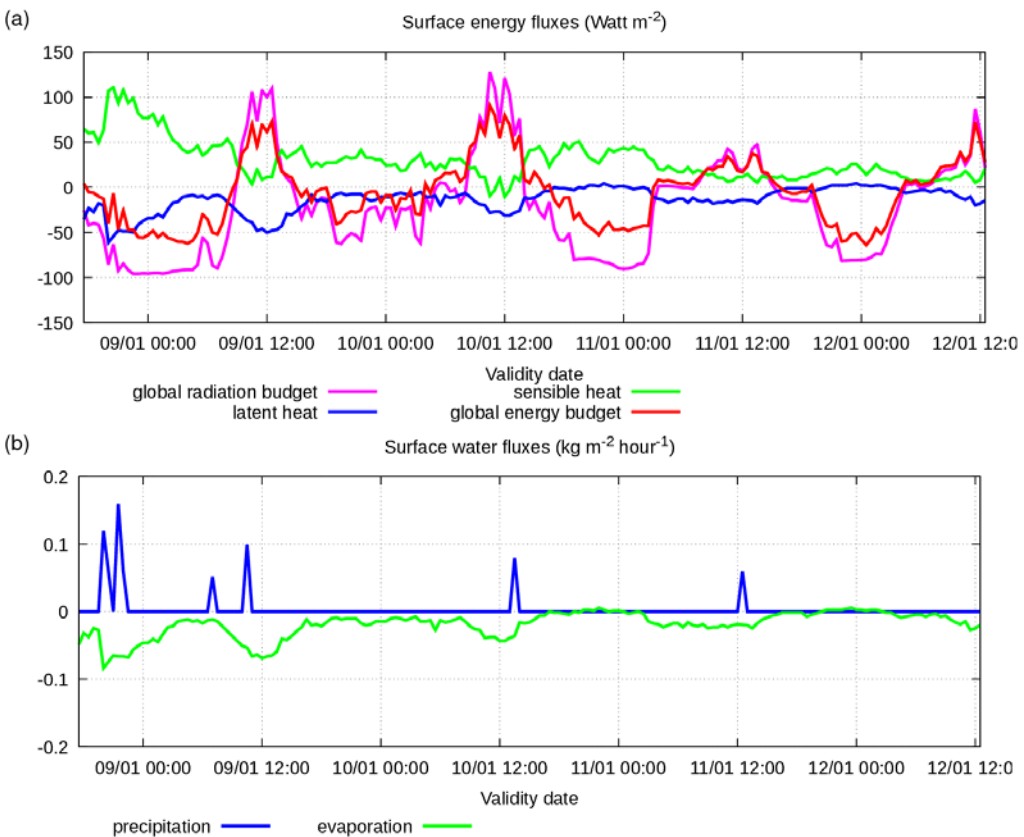

**Figure A4. Observation data (BALTEX) in Falkenberg 8-12 January 2005, parameters at the surface: (a) solar radiation and heat fluxes (W m$^{-2}$); (b) precipitation and evaporation rate (kg m$^{-2}$ hour$^{-1}$).**

This January period is characterised by intensive net radiation flux during the first and second days with significant daytime heating and night-time cooling, accompanied by weak sensible and latent heat fluxes except for the first night, when a significant sensible heat flux is observed; latent heat flux is significant only after light rain in the first day. Some episodes of very light precipitation occur. The evaporation flux has small values.

The simulated soil and moisture data at same depth levels are presented in figures A5 and A6 together with observed data.


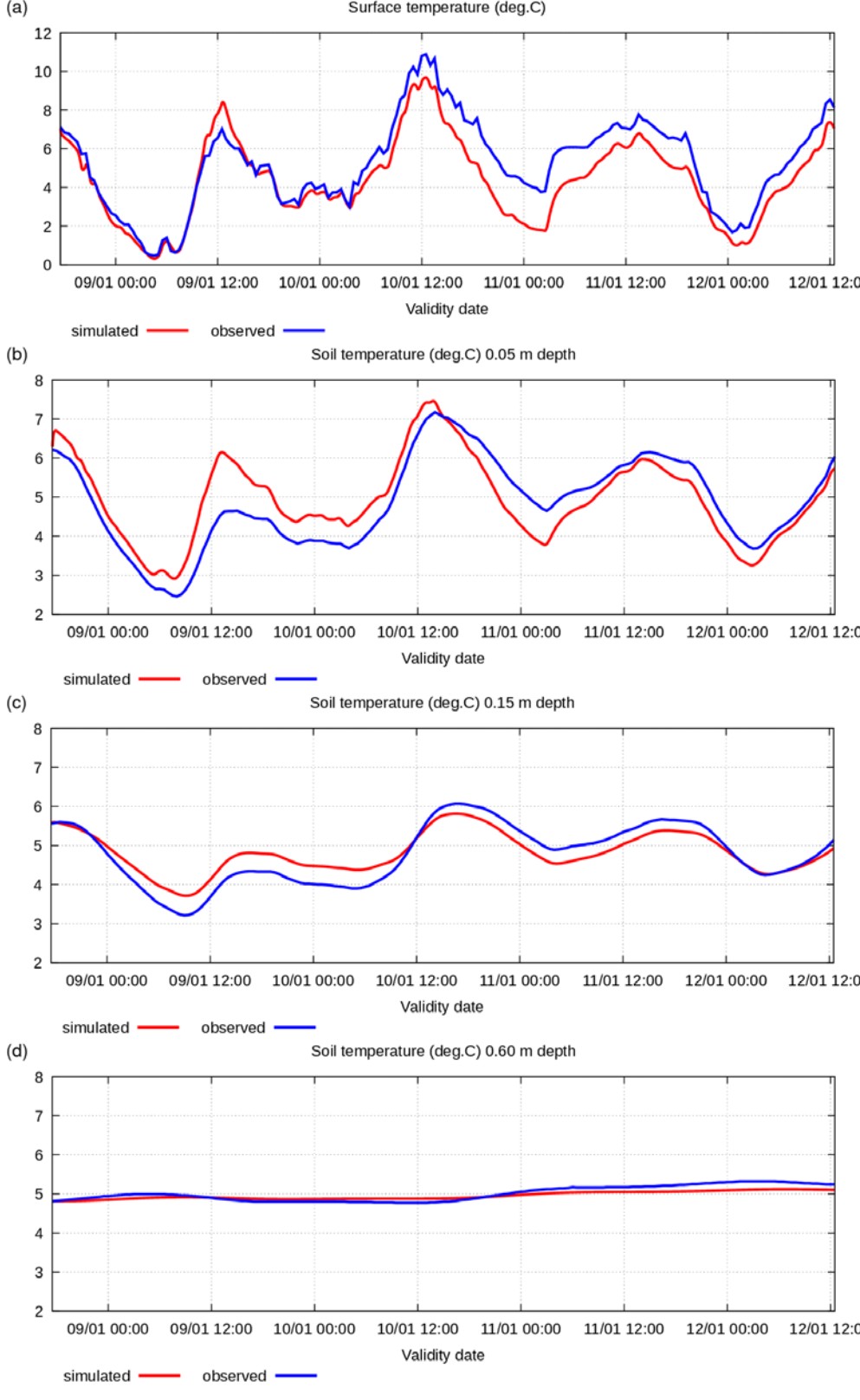

**Figure A5. Simulated and observed 8-12 January 2005 soil temperature (C degree) at the surface and various depths: (a) surface; (b) -0.05 m; (c) -0.15 m; (d) -0.60 m.**

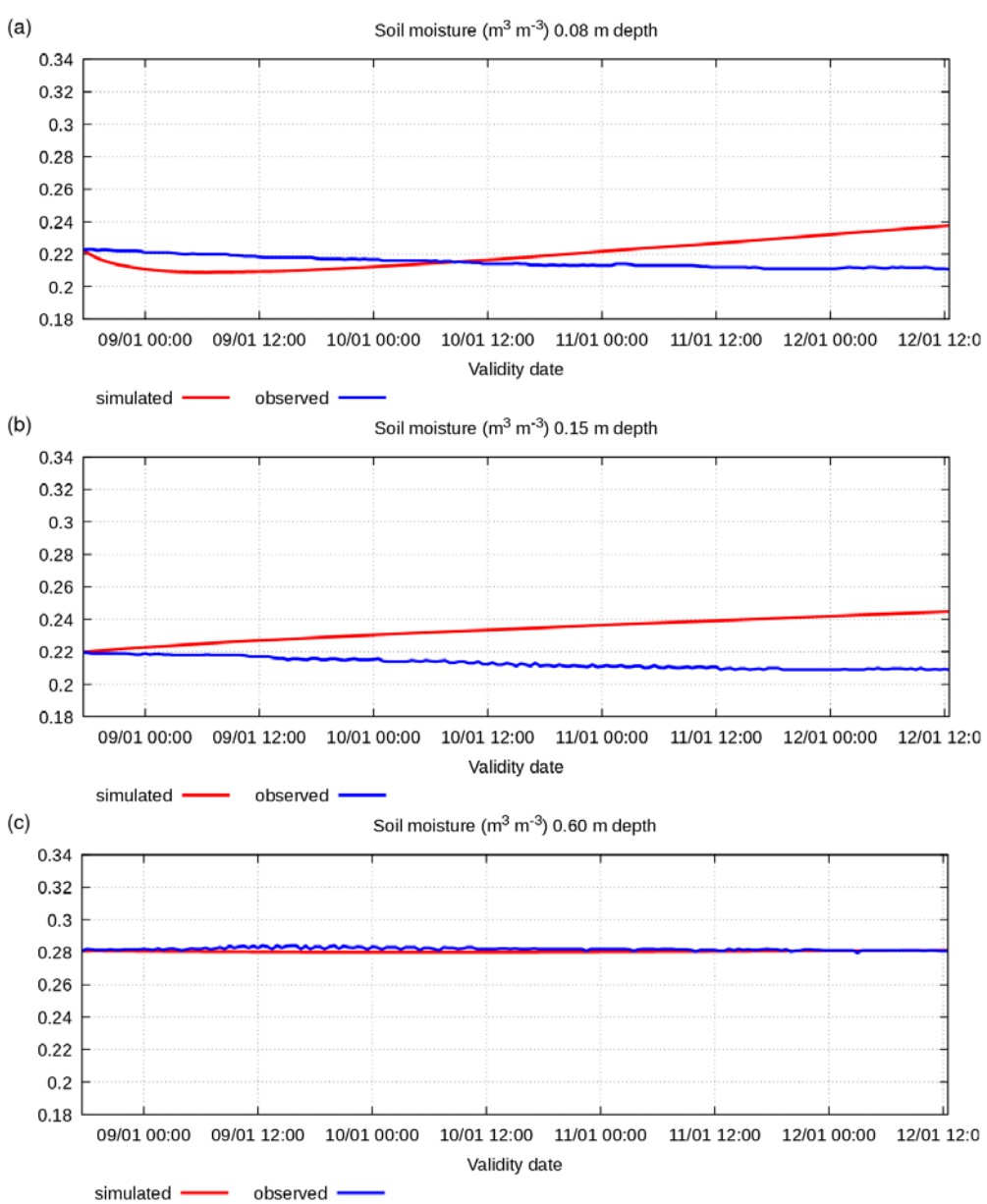


**Figure A6. Simulated and observed 8-12 January 2005 soil specific volumetric water content (m3 m-3) at various depths: (a) -0.08 m; (b) -0.15 m; (c) -0.60 m.**

In this winter case the initial condition for the soil temperature has a very small vertical gradient. For this reason, the simulated

results do not show numerical unbalance. The simulated soil temperature is very close to observed values. At the same time, the initial condition for soil moisture has a significant vertical gradient following observed data. Thus, the simulated soil moisture has the characteristics of numerical disequilibrium, similarly to the summer case. However, in this case the observed moisture has an odd behaviour, it remains almost constant at all vertical levels. The observed soil moisture data indicate that water conductivity is close to zero, but it does not correspond to the conductivity value declared for the observation point (see

table A1). Probably, there are some measurement inaccuracies.

To summarise, it can be said that Pochva scheme is able to simulate the soil temperature and moisture enough correctly. The observed values of soil physical parameters, of forcing scheme parameters and of initial condition parameters have been used in Pochva simulations with no manipulation, and simulations show results close to observation data.

The second type of Pochva simulation has been performed for the period 03/07/2007 03:00 – 31/12/2009 23:30 (912.85 days),

which includes almost 3 years. The sensible heat, latent heat and water vapour fluxes at the surface have not been taken from measurements but simulated by an air surface layer process model based on Monin-Obukhov similarity theory (Monin and Obukhov, 1955) using wind, air temperature and air humidity observations at 40 m above the surface and at the surface. The simulation results and their comparison with the observation data are presented below.

Input data on energy fluxes at the surface and water mass fluxes are shown in figure A7, the simulated soil temperature and

moisture data at same depth levels are shown in figures A8 and A9 together with observed data.

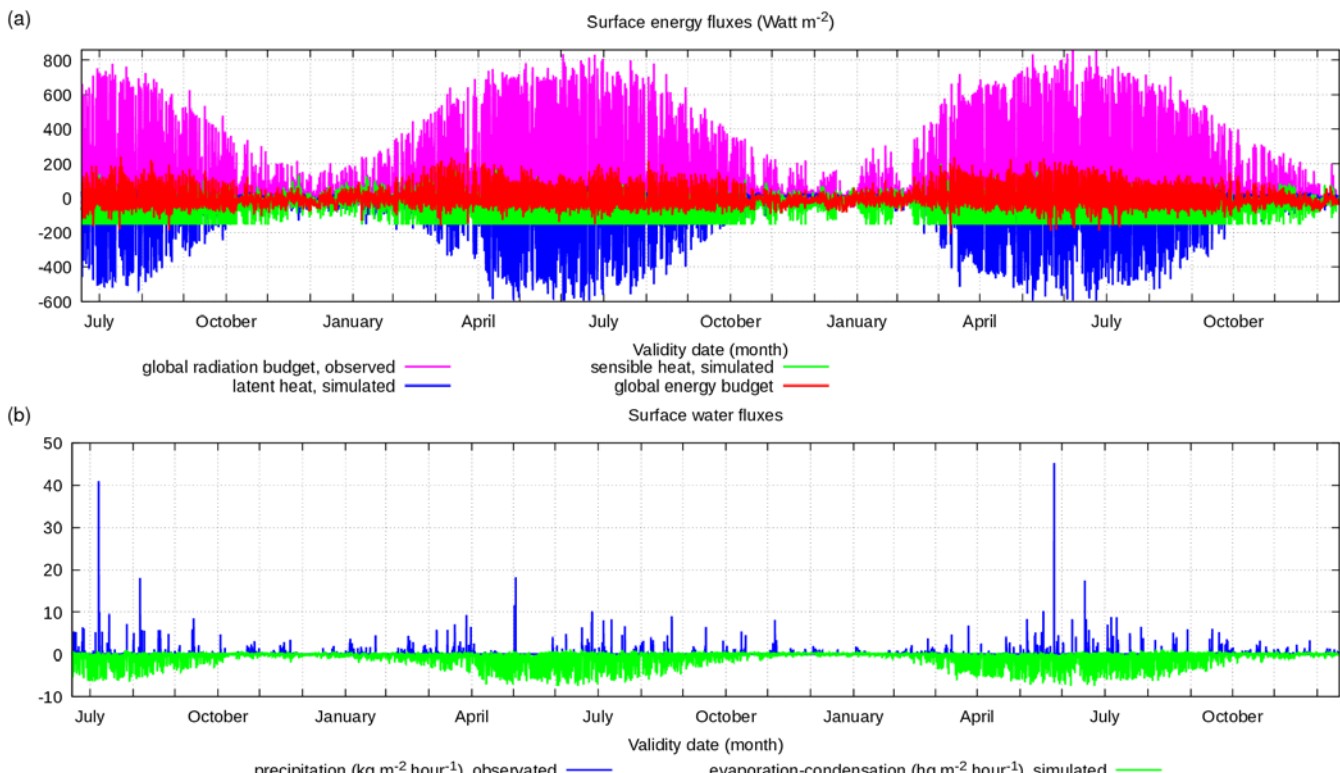

**Figure A7. Simulation and observation data for the period 3 July 2007 – 31 December 2009, parameters at the surface: (a) solar radiation (W m⁻²) observed and heat fluxes (W m⁻²) simulated; (b) precipitation rate (kg m⁻² hour⁻¹) observed and evaporation rate (hg m⁻² hour⁻¹) simulated**


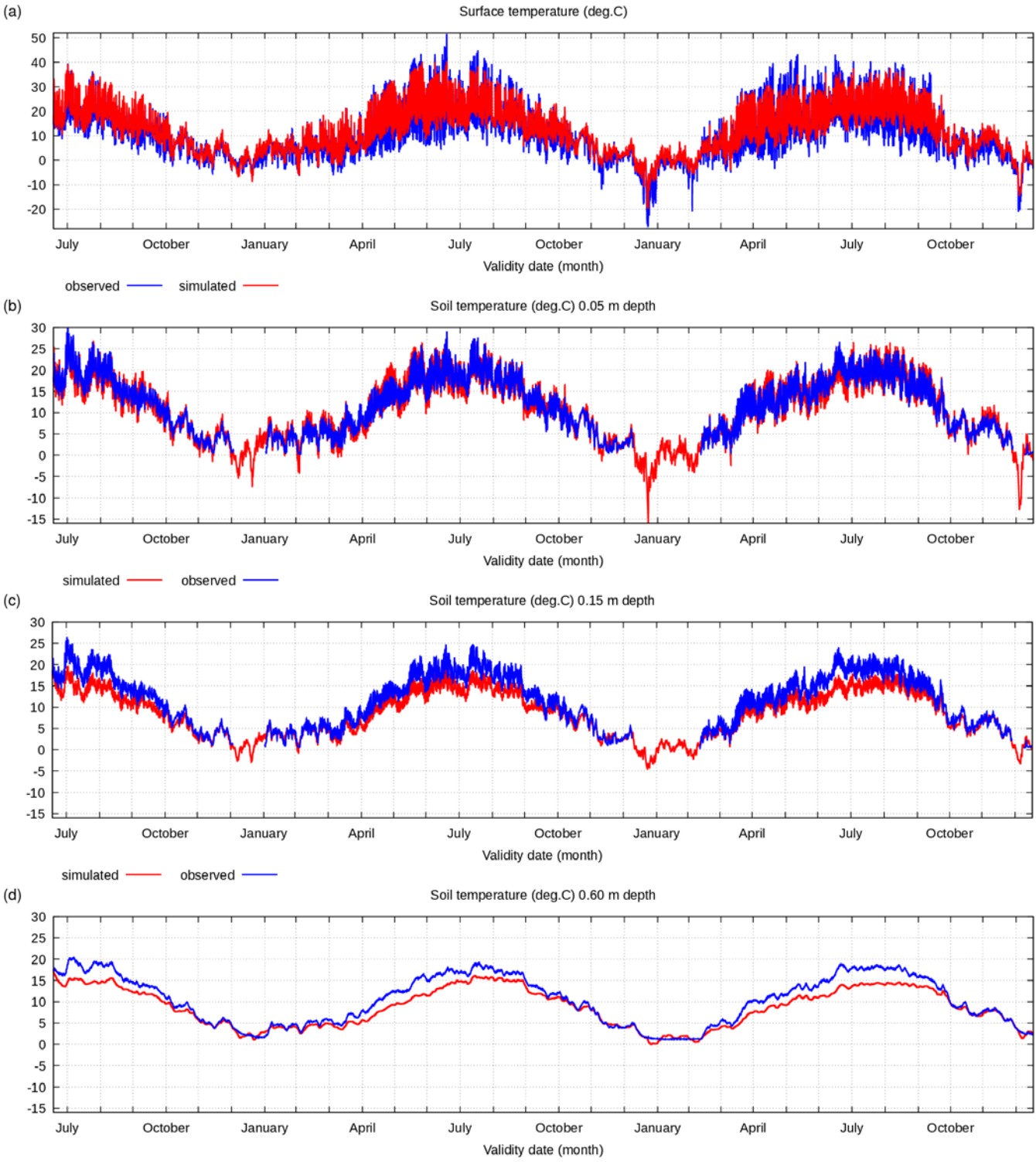

**Figure A8. Simulated and observed soil temperature (°C) at the surface and at various depths: (a) surface; (b) -0.05 m; (c) -0.15 m; (d) -0.60 m for the period 3 July 2007 – 31 December 2009.**

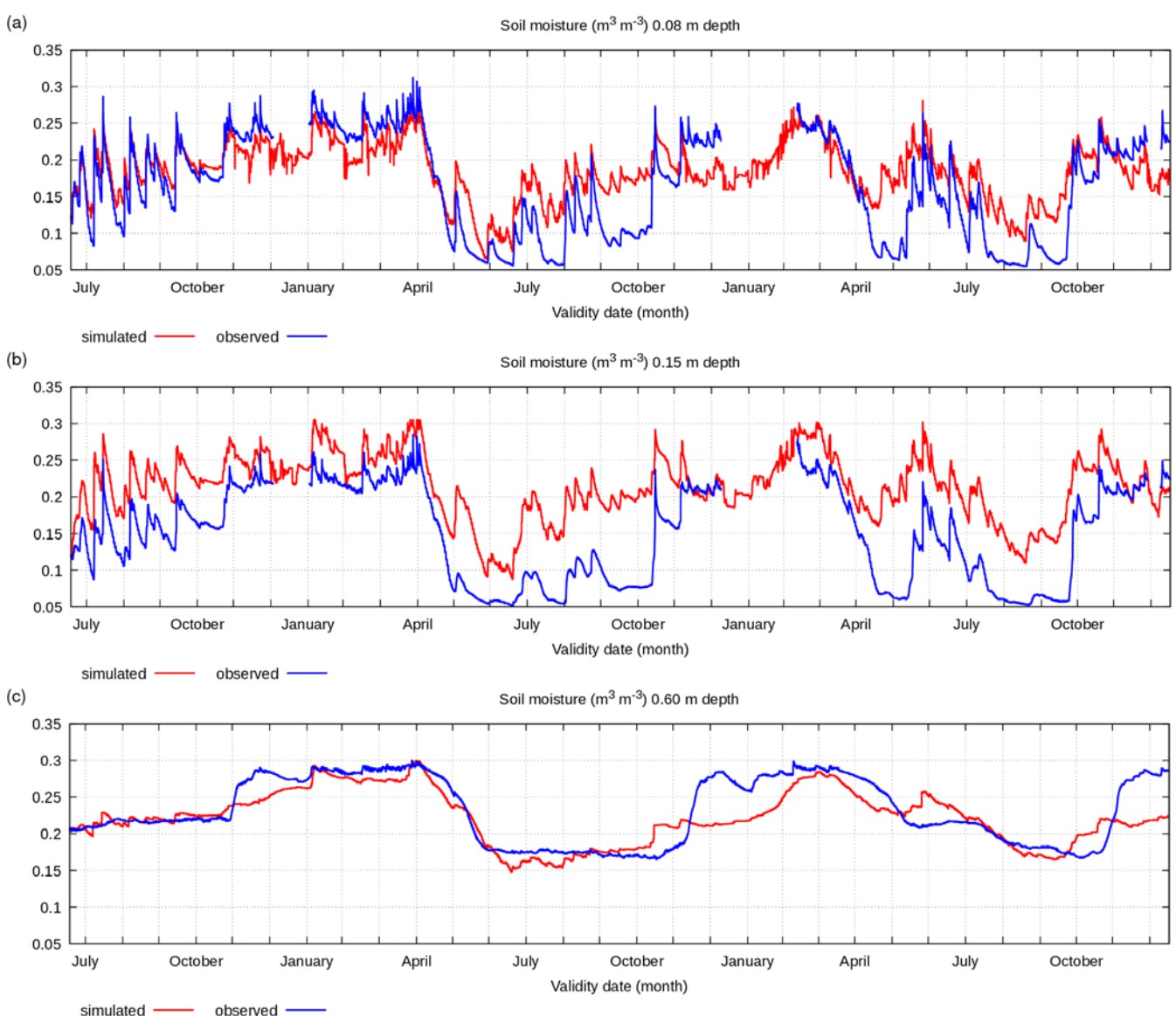


**Figure A9. Simulated and observed soil specific volumetric water content (m3 m-3) at various depths: (a) -0.08 m; (b) -0.15 m; (c) -0.60 m for the period 3 July 2007 – 31 December 2009.**

Figures A7-A9 show a well-defined seasonal cycle of the surface and soil parameters. Periods with very hot and with very

cold weather, as well as periods with precipitation, are present in the observed data. As it can be seen in figures A8 and A9,

unfortunately, at temperatures below 0°C, the soil temperature and soil water content sensors did not operate in both winters. However, the surface temperature sensor operated during the whole simulation period.

Figures A8 and A9 show that the simulated values for various parameters are close to the observed values. The surface temperature is very close to the observed values (fig. A8a), this means that Pochva and the air surface layer schemes approximate the energy flux surface equilibrium very well. This is very important because the main task of the Pochva scheme is to correctly simulate the thermal state of the surface for use in a numerical atmosphere model. The simulation of the soil temperature in the middle levels has some problems with underestimation (by 3-5 degrees) in the summer months (figs. A8b, A8c), this error is also noticeable in the deep layer (fig. A8d), but less pronounced (2-3 degrees). During winter months, when the surface temperature is mostly between 0 and -20 °C, the simulation of soil temperature is correct at all soil levels. The soil water content in the upper layer is mostly correctly simulated (fig. A9a), except for August 2008 and 2009 and October 2008 and 2009 when soil moisture at 8 cm depth was overestimated. At the same time, the soil water content in the deeper layer was simulated correctly (fig. A9c).

In summary, it can be noted that Pochva scheme simulates correctly the temperature and water content of the soil in the upper and lower soil layers in all seasons, but in the middle layer only in the cold season, while in the warm season it underestimates the temperature of this layer and overestimates the water content.

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
