# Peer review of ""Pochva": a new hydro-thermal process model in soil, snow, vegetation for application in atmosphere numerical models"

_Geoscientific Model Development, 2024_

## Author Comment (AC2)

Bare soil

Low vegetation

Liquid water on the plant leaf surface

Snow cover

---

## Author Response (AR2)

**RC1**: 'Comment on gmd-2024-138', Anonymous Referee #1, 19 Sep 2024

This article presents a comprehensive model of hydro-thermal processes in vegetated soil and snow cover, emphasizing the vertical variations of physical parameters along the soil profile. The model features a sophisticated snow processes scheme capable of handling extremely thin and thick snow layers, ensuring accurate simulation of water phase transitions in soil and snow. The interaction between the atmosphere and land surface is simulated, incorporating vegetation and snow cover as independent thermodynamic entities. The model's verification in the NWP model Bolam demonstrates its performance across various climatic zones, revealing biases and RMSEs for temperature and humidity, with particular attention to seasonal variations and geographical differences. The model is based on the idea of describing the thermal state of soil environment using entropy, and pays attention to the soil water phase transition, making it novel and potentially applicable in atmospheric and climate research. However, the article still needs some revisions before it is suitable for publication.

1.
(1) The colors in Figure 1 seem to be inconsistent with the colors in the legend (color for "Liquid water on the plant leaf surface"). Please check if there is a problem.
(2) The color of the water zone of "Liquid water on the plant leaf surface" in the figure and in the legend is the same, but it is semi-transparent, in the figure the color of the vegetation zone shows through it. I aimed to make the effect that one zone is superimposed on another, i.e. it is a part of it, not a separate zone, but I am agree than it is not very clear.
(3) The figure 1 was redrawn.

2.
(1) The meaning of $\beta$ veg in equation 11 does not seem to be given in the article.
(2) There was a mistake after formula (6), a and b instead of $\alpha$ and $\beta$, formula (11) is the explanation of $\beta$ coefficient from (6), I hope now it is clear.
(3) Line 153 in corrected manuscript.

3.

(1) In Figure 2, it mentions k=nlev_soil, where nlev_soil refers to the maximum number of soil layers?Can this maximum number of layers vary, or is it a fixed value? Also, is the vertical stratification of soil layers in the model uniform, or does it vary across different spatial locations?

(2) Yes, you are right, k=nlev_soil is the index of the bottom level in soil column, the boundary condition for solving the differential equation is defined at this level.
Your questions about the distribution of vertical levels and the depth of the bottom level are correct. I did not put the answers to these questions in the text because it is a special question that requires an extensive discussion. Vertical levels in soil for numerical solving the equations of heat and moisture exchange in soil do not have an uniform distribution, the soil layers thickness increases with depth according to an exponential law. But, of course, this distribution can be changed. The definition of the bottom level depth depends on how the model will be applied for specific purposes. The presented numerical scheme assumes that the depth of the bottom level for solving the heat exchange equation

may not coincide with the depth of the bottom level for solving the moisture exchange equation. In both cases, this depth is determined by the boundary conditions to be specified. Also, the scheme assumes that both of these depths may be different at different domain numerical grid point depending on the geographical conditions that define the boundary conditions for the heat and moisture exchange equations. I plan to publish my study with a proposal on how these conditions can be set in different geographical conditions, as it is quite a big special study.

(3) The author included a brief explanation of both points in the text, lines 179-183 in corrected manuscript.

4.

(1) Why were the simulation results from 2013 to 2015 over the European region selected for validating the Pochva model, instead of selecting periods from the routine operational weather prediction results of CNR-ISAC, where Pochva has already been applied?

(2) That's a correct question. The verification of the application of the "Pochva" scheme presented in this paper was done before "Pochva" was introduced into the versions of the NWP models used in CNR-ISAC for operational forecasting, moreover the operational setup of the models was not ideal for verifying the Pochva scheme, these are the reasons for which a dedicated long test experiment was performed. The NWP model chosen for this test was the Bolam model, a hydrostatic numerical model that can have an extensive domain. The author aimed to test the Pochva scheme over a wide range of geographical conditions, i.e. a wide domain. Using of the global model "Globo" was not reasonable, since the global domain assumes that 3/4 of the domain is covered by the ocean, i.e. not useful for soil scheme testing, but increases computational resource consumption. The Bolam model was applied on a domain much more extensive than that used for operational forecasting to capture the hot desert regions of Western Asia, the oceanic regions of Europe, whole Mediterranean region, whole Scandinavia (marine polar climate) and the region up to the Northern Urals (continental polar climate).

(3) Nothing changed.

5.

(1) In the evaluation of land surface models, offline mode is commonly used. Can the Pochva model perform offline simulations? If so, it is suggested to use offline simulations for evaluation to more effectively avoid the uncertainty in atmospheric models from influencing the evaluation results.

(2) You are right, it would be correct to test the soil scheme in the column version. Of course, the scheme, in its current form, can be used this way. The author presented 6 test datasets obtained from the simulation with the full NWP model for this application for testing the model. The author performed verification of offline column version of Pochva with using of observation data. The results of this verification was included in the manuscript.

(3) The author has written a new part of manuscript, the appendix in the corrected manuscript (lines 845-943), with presentation of results of Pochva simulation in off-line column version using observation data forcing and with the verification of obtained results with observation data.

Minor modifications:

1.

(1) It is recommended to remove the phrase 'of National Center of Atmospheric Research (USA)' after the Common Land model in lines 26-27, as the later development and application of this model were mostly carried out in China.
(2) Thank you for the correction.
(3) Correction made, line 27 in the corrected manuscript.

2.
(1) line 316 then shuld be than?
(2) Yes.
(3) Correction made, line 341 in the corrected manuscript.

3.
(1) line 385 does the equation in this line need a number?
(2) No, this equation is trivial clarification.
(3) The equation has been included in the text, line 409 in the corrected manuscript.

4.
(1) line 559 snow us should be snow was?
(2) Yes.
(3) Correction made, line 584 in the corrected manuscript.

Finally, the reviewer would like to express his respect for the author of the article. As we all know, developing a new land surface model all by oneself is extremely difficult and may face various challenges.

**RC3**: ['Comment on gmd-2024-138'](), Anonymous Referee #2, 24 Oct 2024

General Comment

The paper presents "Pochva", a model of hydro-thermal processes at the Earth surface to be included in meteorological or climate models. The paper is interesting and can provide a valuable contribution to the community. However, in my opinion, some weaknesses should be solved before the paper can be accepted for publication.
In most cases, it is not clear what are the elements of novelty of this scheme with respect to other land surface models. The Author should better emphasize the similarities and the differences of the new parameterization with respect to existing land surface models, also to highlight why users should choose "Pochva" against other land surface models.
The analysis of the performance of the parameterization presented in the paper does not allow a clear evaluation of the strengths and weaknesses of the scheme, since Pochva is online coupled with the NWP model Bolam and errors in an NWP model can be caused by different interacting factors. It would be better to evaluate Pochva offline (directly forced by observations). Otherwise, it would be

interesting to compare the results of Bolam online coupled with different land surface models. In this way, it would be easier to evaluate the strengths and weaknesses of Pochva.

Major Comments

Model description
(1) Model description: the formulations adopted to describe the various physical processes should be better justified, pointing out if they have been taken from previous literature, adapted for this new scheme or if they are completely original. As an example, in Section 2.2, it is unclear how and why Equations 3, 4, and 5 were chosen for defining the empirical soil coefficient $\alpha_{soil}$. This is valid for most of the equations presented in this paper. The Author should try to emphasize the elements of novelty of Pochva through comparison with the formulations used in other land surface models. In the present version of the manuscript, the comparison with other existing land surface models is completely absent, whereas Pochva should be described in the framework of the literature in this field.
(2) Author has rewritten the explanation to the presented formulas in order to more clearly indicate which formulas are completely original and which are taken from the literature and adapted in this paper so that the results of application of the soil scheme in a full atmospheric model would give the best results.
(3) Author has made additions to the manuscript: lines 61, 122-125, 144-145, 154, 159, 166-168, 979-980, 992-993, in the corrected manuscript.

Model evaluation
(1) Model evaluation: as written in the General Comment, in the present version of the manuscript it is difficult to evaluate the performance of Pochva because (i) Bolam model errors can be due to different reasons, so it is not clear if we are analysing errors related to the land surface model or to other components of the NWP model, (ii) a comparison with the results provided by other land surface models is not present. Therefore, it is not clear how to evaluate strengths and weaknesses of Pochva. Indeed, in Section 7 the Author speculates on the possible causes of model errors, but clear conclusions cannot be drawn.
(2) This comment is fair. The main purpose of the manuscript is to present a surface-soil-vegetation-snow cover model as part (a parameterisation) of a numerical model of atmospheric processes. The problem is that the scheme of parameterisation of physical processes is developed and tested in the framework of a specific complete numerical atmospheric model with the purpose of improving the parameterisation of these physical processes in the model. "Pochva" has been tested in NWP model "Bolam" for the reasons presented in chapter 6. The model Bolam had previously used a different, simpler land processes parametrization. It would be possible to present the comparison of Pochva and the previous land processes scheme in Bolam, but, unfortunately, that previous scheme has never been presented in publications. At the same time, the insertion of Pochva scheme into some other NWP model is a very time-consuming task that requires a special great work, which is beyond the scope of the presented work. Following the comment, author has performed the evaluation of Pochva scheme using observation data. Author has found a freely available observation dataset containing the data about solar radiation fluxes, sensible and latent heat fluxes at the surface, about the surface temperature

and the soil temperature at various soil depths, about soil moisture at various soil depths. This dataset includes the information about soil and vegetation physical parameters in the point of observation. Author obtained observation data, decoded them, found all data and time when observation data of all forcing parameters are present, searched for periods when time gap between 2 consecutive observations was no more than 2 hours, and searched for the longest among periods found, performed a time interpolation of observation data to obtain forcing parameter data with a time step useful for a numerical simulation (1 minute). The author prepared initial condition data and forcing data for simulation with column version of Pochva, and determined soil and vegetation parameters used in Pochva following information available in the dataset description. The author performed numerical simulation with Pochva for 2 chosen cases (summer and winter), verified the obtained simulation results with observation data from the same dataset and formulated the verification results.

(3) The author has written a new part of manuscript, the appendix in the corrected manuscript (lines 845-943,) presenting results of Pochva simulation in off-line column version using observation data forcing and showing the verification of obtained results with respect to observation data.

Quality of the presentation
(1) Quality of the presentation: The quality of the presentation is not always satisfying. A review by a native English speaker would be beneficial.
(2) This requirement is difficult to fulfill.
(3) The author has endeavoured to correct the manuscript with the help of colleagues and application of an automatic corrector service.

Minor and technical comments

(1) Line 30: check the reference style
(2) Correction made.
(3) Correction made, lines 30, 1004-1005 in the corrected manuscript.

(1) Line 43: "to the inhomogeneity IN soil parameters along…"
(2) Correction made.
(3) Correction made, line 43 in the corrected manuscript.

(1) Line 67: "is composed OF a set of fractions…"
(2) Correction made.
(3) Correction made, line 68 in the corrected manuscript.

(1) Lines 86-87: from this sentence it seems that the fraction of wet low-vegetation leaves is pre-assigned, whereas it is variable and calculated by the parameterization.
(2) The clarification is made.
(3) Correction made, lines 96-98 in the corrected manuscript.

(1) Equation 2: $T_0$ should be defined.

(2) Yes.
(3) Definition made, line 120 in the corrected manuscript.

(1) Line 108 and 110: asoil should be $\alpha_{soil}$

(2) Correction made.
(3) Correction made, lines 119 and 121 in the corrected manuscript.

(1) Line 112: "see section 6" should be "see section 7"
(2) Correction made.
(3) Correction made, line 125 in the corrected manuscript.

(1) Line 137 and line 148: bveg should be $\beta_{veg}$

(2) Correction made.
(3) Correction made, lines 153 and 169  in the corrected manuscript.

(1) Line 146: $W\ m^{-2}$ and not $Watt\ m^{-2}$
(2) Correction made.
(3) Correction made, line 164 in the corrected manuscript.

(1) Line 158: it is not clear how $k_{root}$ is evaluated.

(2) Clarification made.
(3) Clarification made, lines 179-183 in the corrected manuscript.

Line 209: $W\ m^{-2}$ and not $Watt\ m^{-2}$
(2) Correction made.
(3) Correction made, line 234 in the corrected manuscript.

(1) Line 210: should be
(2) I do not understand this remark.
(3) No modification.

(1) Line 252: "can either be composed OF soil partially covered…"
(2) Correction made.
(3) Correction made, line 277 in the corrected manuscript.

(1) Lines 400 and 453: "Let's" is colloquial.
(2) Correction made.
(3) Correction made, lines 424 and 477 in the corrected manuscript.

(1) Equation 68: $f_{ice}{}^{soil}$ should be $f_b$

(2) Yes, thank you, correction made.
(3) Correction made, line 474 in the corrected manuscript.

(1) Line 458: "temperature before and after…"
(2) Correction made.
(3) Correction made, line 4832in the corrected manuscript.

(1) Line 500: "and density are conserved."
(2) Correction made.
(3) Correction made, line 524 in the corrected manuscript.

(1) Line 516: errors in displaying the symbols.
(2) Correction made.
(3) Correction made, line 540 in the corrected manuscript.

(1) Lines 517-521: I do not understand the hypothesis that all the liquid water at the beginning of the time step is at level $k$ and at the end of the time step is at level $k+1$, also considering that the layer thickness is variable in time. This assumption should be better justified.
(2) Clarification made.
(3) Clarification inserted, lines 545-546 in the corrected manuscript.

(1) Line 549: rsnow should be $\rho_{snow}$.
(2) Correction made.
(3) Correction made, line 574 in the corrected manuscript.

(1) Line 559: "The heat conductivity of snow IS defined…"
(2) Correction made.
(3) Correction made, line 585 in the corrected manuscript.

(1) Line 559: check the correct reference style.
(2) Correction made.
(3) Correction made, line 5854in the corrected manuscript.

(1) Line 567: Is $S_{snow}$ the entropy of soil water?
(2) No, it's my error, thank you. Corrections made.
(3) Correction made, lines 592 in the corrected manuscript.

(1) Line 580: soil water entropy or snow entropy?
(2) No, it's my error, thank you. Correction made.
(3) Correction made, line 605 in the corrected manuscript.

(1) Lines 593-594: "diagnose the length of the time interval during which snow is exposed to melting".

(2) Correction made.
(3) Correction made, line 619 in the corrected manuscript.

(1) Line 610: "at 2 m above SURFACE"
(2) Correction made.
(3) Correction made, line 636 in the corrected manuscript.

(1) Line 655: "it can thus be noticed that…"
(2) Correction made.
(3) Correction made, line 682 in the corrected manuscript.

(1) Line 747: "2-m dew-point temperature"
(2) Correction made.
(3) Correction made, line 779 in the corrected manuscript.

(1) Line 762: "winter and autumn" should be "spring and autumn".
(2) Correction made.
(3) Correction made, line 794 in the corrected manuscript.

(1) Lines 782-783: The Author says that the presented model can be useful for modelling conditions over polar or cold climate zones and in the forecast of snow cover. However, model results show the highest errors in winter in continental and mountain areas. In this regard, the Author says that the higher errors can be explained by a poor simulation of the snow cover. Therefore, the conclusions seem inconsistent with the results.
(2) It's good remark. I supplemented the analysis of simulated results for cold climate area.
(3) Additions made, lines 718-720  in the corrected manuscript.

(1) Line 785: "or to OBSERVATIONS of air surface…"
(2) Correction made.
(3) Correction made, line 814 in the corrected manuscript.

(1) Figure 1: the color of "Liquid water on the plant leaf surface" is not consistent between figure and legend.
(2) The color of the water zone of "Liquid water on the plant leaf surface" in the figure and in the legend is the same, but it is semi-transparent, in the figure the color of the vegetation zone shows through it. I aimed to make the effect that one zone is superimposed on another, i.e. it is a part of it, not a separate zone, but I am agree than it is not very clear.
(3) The figure 1 was redrawn.

(1) Figures 4-7: I would plot the orography in gray shading to better see the station points.
(2) This is good remark, thank you for the advice.
(3) The figure 4-7 were redrawn.

(1) Caption Figure 9: "Average diurnal cycle…"
(2) I do not agree, because it would be a redundant definition, BIAS is the arithmetic average of error.
(3) No modification.

(1) Caption Figure 10: "Same as figure 9…"
(2) Correction made.
(3) Correction made, line 764 in the corrected manuscript.

(1) Caption Figure 11: "Average diurnal cycle…"
(2) I do not agree, because of it shroud be a redundant definition, BIAS is the arithmetic average error.
(3) No modification.

(1) Caption Figure 10: "Same as figure 11…"
(2) Correction made.
(3) Correction made, line 803 in the corrected manuscript.

**The editorial support team** 2 Dec 2024

(1) Please note that your reference list has not been compiled according to our
standards. Please consider adjusting your reference list with the next revision of
your manuscript. The manuscript preparation guidelines can be seen at:
[https://www.geoscientific-model-development.net/for_authors/manuscript_preparation.html](https://www.geoscientific-model-development.net/for_authors/manuscript_preparation.html).
(2) Correction made.
(3) Correction made, lines 945-1010 in the corrected manuscript.

(1) If possible please improve the resolution of figures 4 to 7.
(2) Yes, it is possible.
(3) Correction made.

(1)The appendix figures and tables must be relabelled: Figure 13 --> Figure A1; Figure
14 --> Figure A2, Table 1 --> Table A1, etc. Please ensure to use the correct labels
also in the text.
(2) Correction made.
(3) Correction made, lines 845-943 in the corrected manuscript.

---

## Author Response (AR3)

**Report #2**
**Submitted on 06 Jan 2025**
**Anonymous referee #2**

Suggestions for revision or reasons for rejection

(visible to the public if the article is accepted and published)

General Comment

The Author only partly addressed my previous major comments, concerning in particular model evaluation and quality of the presentation.

Major Comments

(1) Model evaluation: the Author added, in the Appendix, an evaluation of the model in offline mode (i.e., directly forced by observations). However, the comparison is too short to obtain reliable information on the model performance. Moreover, as specified by the Author "A numerical scheme needs a warm-up period in order to reach numerical equilibrium after initialisation. Unfortunately, it has not been possible to find a continuous observation period of sufficient length in order for the simulated parameters to reach a balanced state" and "Probably, there are some measurement inaccuracies". So, it seems that the dataset used to force and verify the model is not suitable for this scope. It is then strongly recommended to use a more suitable dataset for this scope, evaluating model results for a longer time period, similarly to the online evaluation presented in Section 7.

(2) I have been performed another verification study. The study is based on the continued simulation of 912 days (2,5 years) with the column version of the "Pochva" scheme. The observed data on the radiation budget and precipitation flux at the surface, the computed data on sensible heat and latent heat fluxes, as well as the evaporation flux in the air surface layer have been used as atmospheric forcing parameters in the simulation. The computation of the fluxes in the surface layer has been performed using the observed data on wind, pressure, temperature and humidity at the surface and at 40 m above the surface. The bottom conditions for the soil temperature and the soil water content have been defined using the observed values at the deepest levels available. The initial conditions for indicated prognostic variables have been defined using the measured values at the various soil levels. The simulation was successful and provided good results. The simulated values of the soil prognostic variables have been compared with the observed values. The results of the performed verification are presented in the Appendix of the manuscript.

(3) The obtained results of the new verification studies are presented in the lines 897-900, 964-1000 in the corrected manuscript.

(1) Quality of the presentation: The quality of the presentation is still not satisfying. Actually, little has changed with respect to the previous version of the manuscript.

(2) I am sorry that the English language style of the manuscript is not literary and expressive enough. I have re-checked everything thoroughly again and have corrected the manuscript in many places. I hope that the text clearly expresses the intended meaning.

(3) Many corrections of the text have been made throughout the whole paper.

Minor and technical comments

(1) The reference style is still wrong throughout the manuscript.

(2) The reference style of the manuscript has been adapted to the requirements of the journal.

(3) References in the manuscript.

(1) Line 21: "current model" or "current models"?

(2) Yes

(3) Correction made, line 21 in the corrected manuscript.

(1) Line 58: delete comma.

(2) No, this comma is important for the following scheme presentation.

(3) No modification.

(1) Lines 64-65: the meaning of this sentence is not clear.

(2) The sentence has been rewritten.

(3) Correction made, lines 64-66 in the corrected manuscript.

(1) Line 97: "is determined by THE RATIO BETWEEN THE water mass deposited on the leaf surface and THE maximum value of this mass, which is also EVALUATED using A dataset."

(2) The sentence has been rewritten.

(3) Correction made, lines 97-98 in the corrected manuscript.

(1) Line 115: what is $q_{vatm1}$?

(2) The definition has been corrected.

(3) Correction made, line 116 in the corrected manuscript.

(1) Line 116: (kg kg$^{-1}$), superscripts in units should be used (throughout the manuscript).

(2) Yes.

(3) Correction made, line 116 in the corrected manuscript.

(1) Line 121: \alpha$_{soil}$, "soil" should be a subscript.

(2) Yes.

(3) Correction made, line 121 in the corrected manuscript.

(1) Line 136: "temperature at 2 m…"

(2) Yes.

(3) Correction made, line 136 in the corrected manuscript.

(1) Lines 154-155: please reformulate this sentence.

(2) The sentence has been rewritten.

(3) Correction made, lines 155-156 in the corrected manuscript.

(1) Line 180: "root zone depth is defined using a suitable vegetation dataset" should be in a separate sentence.

(2) Yes.

(3) Correction made, line 181 in the corrected manuscript.

(1) Equations 11-13 are not introduced in the text.

(2) Yes.

(3) Correction made, lines 153-154 in the corrected manuscript.

(1) Equation 16: can the snow surface temperature exceed 0 °C?

(2) No, it is physically impossible, but it is possible in a numerical algorithm.

(3) No modification.

(1) Equation 35: what is $\Phi_{v\ surf}^{turb}$?

(2) The equation has been corrected, thank you.

(3) Correction of the equation (35) made.

(1) Line 372: "is THE pressure gradient…"

(2) Yes.

(3) Correction made, line 374 in the corrected manuscript.

(1) Line 427: $S_{soil}$

(2) Yes.

(3) Correction made, line 429 in the corrected manuscript.

(1) Lines 463-464: refer to the relevant equation.

(2) The sentence has been rewritten.

(3) Correction made, lines 465-467 in the corrected manuscript.

(1) Lines 535-536: this sentence is not clear: it seems that variations of the solid component take place only in the presence of solid precipitation or sublimation/deposition of water vapour. However, variations of the solid component can also occur due to melting in any snow layer.

(2) Yes, this sentence was confusing, it has been rewritten.

(3) Correction made, lines 539-540 in the corrected manuscript.

(1) Lines 541-546: I still do not understand this hypothesis. Why cannot the water drain through more than one layer in one time step?

(2) Since an exact representation of the vertical drainage speed is not crucial for a correct simulation of the dynamics of the snow layer, for simplicity in numerical solution it has been decided to let the water flow through exactly one layer during one time step.

(3) No modification.

(1) Line 550: $\Phi_{m kh}$

(2) Yes

(3) Correction made, line 552 in the corrected manuscript.

(1) Lines 580-581: $\rho_{snow}^{fresh}$ and $\rho_{snow}^{old}$

(2) Yes

(3) Correction made, line 584 in the corrected manuscript.

(1) Line 597: "respectively" is repeated.

(2) Yes, the text has been corrected.

(3) Correction made, line 600 in the corrected manuscript.

(1) Line 599: "The values of the physical parameters at half levels ARE computed".

(2) Yes.

(3) Correction made, line 602 in the corrected manuscript.

(1) Line 602: "taking INTO account".

(2) Yes.

(3) Correction made, line 605 in the corrected manuscript.

(1) Line 608: k should not be superscript.

(2) Yes, it has been corrected.

(3) Correction made, line 611 in the corrected manuscript.

(1) Line 612: "below or equal to 0 °C".

(2) Yes.

(3) Correction made, line 615 in the corrected manuscript.

(1) Line 662: "experiences bias values between"

(2) Yes.

(3) Correction made, line 665 in the corrected manuscript.

(1) Line 705: "moreover the points…", start a new sentence.

(2) Yes.

(3) Correction made, line 708 in the corrected manuscript.

(1) Line 716: "it is lower…", start a new sentence.

(2) Yes.

(3) Correction made, line 718 in the corrected manuscript.

(1) Line 718: "by a similar COLD climate".

(2) Yes.

(3) Correction made, line 721 in the corrected manuscript.

(1) Line 766: "it is positive during daytime and negative in the evening and morning hours", looking at fig. 9 the behavior seems different.

(2) Yes, the analysis has been corrected, thank you.

(3) Correction made, lines 768-772 in the corrected manuscript.

(1) Line 773: "for 2-m temperature".

(2) Yes.

(3) Correction made, line 775 in the corrected manuscript.

(1) Line 774: "The overall errors are not high", "high" is very qualitative if not compared with typical errors found in other similar studies.

(2) The comment has been clarified.

(3) Correction made, lines 776-777 in the corrected manuscript.

(1) Line 778: full stop is missing.

(2) Yes.

(3) Correction made, line 780 in the corrected manuscript.

(1) Line 784: "while THE worst scores"

(2) Yes.

(3) Correction made, line 786 in the corrected manuscript.

(1) Line 788: "occurs AT daytime"

(2) Yes.

(3) Correction made, line 790 in the corrected manuscript.

(1) Line 796: full stop is missing.

(2) Yes.

(3) Correction made, line 798 in the corrected manuscript.

(1) Lines 813-814: The Author has still not demonstrated that Pochva works better than other models over snow cover or with a stable surface layer.

(2) The verification results presented in the manuscript have shown good scores for air temperature and humidity in the surface layer in the Cold Steep, Subarctic and Mountains climatic zones.

(3) No modification.

(1) Line 866: "for soil water exchange processes…", start a new sentence.

(2) The text has been rewritten

(3) Lines 866-872 in the corrected manuscript.

(1) Figures A1 and A4: what is net? (red line).

(2) The comment has been clarified "global radiation budget".

(3) Figures A1, A4, A7 in the manuscript.

(1) Line 906: "characteristics: a light…"

(2) Yes.

(3) Correction made, line 927 in the corrected manuscript.

(1) Lin 942: "observed values of soil physical parameters, of forcing scheme parameters"

(2) Yes.

(3) Correction made, line 962 in the corrected manuscript.

(1) All Figures: use always the correct unit symbols.

(2) Yes.

(3) All figures and these captions have been controlled and corrected.

---

## Author Response (AR4)

**Report #1**
**Submitted on 26 Feb 2025**
**Anonymous referee #2**

Suggestions for revision or reasons for rejection

(visible to the public if the article is accepted and published)

The Author improved the manuscript with new analysis. However, in my opinion, the presentation quality, considering in particular the appropriate use of the English language, is still not satisfactory. English should be proofread by a native speaker.

Author Response

The manuscript has been corrected by a native English speaker. References in the text have been corrected.